# Tumor stiffening reversion through collagen crosslinking inhibition improves T cell migration and anti-PD-1 treatment

Alba Nicolas-Boluda[1,2,3], Javier Vaquero[4,5,6,7], Lene Vimeux[1,2], Thomas Guilbert[1], Sarah Barrin[1,2], Chahrazade Kantari-Mimoun[1,2], Matteo Ponzo[8], Gilles Renault[1], Piotr Deptula[9], Katarzyna Pogoda[10], Robert Bucki[9], Ilaria Cascone[8], José Courty[8], Laura Fouassier[4], Florence Gazeau[3†]*, Emmanuel Donnadieu[1,2†]*

[1]Institut Cochin, INSERM U1016/CNRS UMR 8104, Université de Paris, Paris, France; [2]Equipe Labellisée Ligue Contre le Cancer, Paris, France; [3]Laboratoire Matière et Systèmes Complexes (MSC), CNRS, Université de Paris, Paris, France; [4]Sorbonne Université, Inserm, Centre de Recherche Saint-Antoine, CRSA, Paris, France; [5]TGF-β and Cancer Group, Oncobell Program, Bellvitge Biomedical Research Institute (IDIBELL), Barcelona, Spain; [6]LPP (Laboratoire de physique des plasmas, UMR 7648), Sorbonne Université, Centre national de la recherche scientifique (CNRS), Ecole Polytechnique, Paris, France; [7]Oncology Program, CIBEREHD, National Biomedical Research Institute on Liver and Gastrointestinal Diseases, Instituto de Salud Carlos III, Barcelona , Spain ; [8]CNRS ERL 9215, CRRET laboratory, University of Paris-Est Créteil (UPEC), Paris, France; [9]Department of Medical Microbiology and Nanobiomedical Engineering, Medical University of Białystok, Białystok, Poland; [10]Institute of Nuclear Physics, Polish Academy of Sciences, Kraków, Poland

*For correspondence:
florence.gazeau@u-paris.fr (FG);
emmanuel.donnadieu@inserm.fr
(ED)

†These authors contributed
equally to this work

Competing interests: The
authors declare that no
competing interests exist.

Reviewing editor: Bernard
Malissen, Centre d'Immunologie
de Marseille-Luminy, Aix
Marseille Université, France

**Abstract** Only a fraction of cancer patients benefits from immune checkpoint inhibitors. This may be partly due to the dense extracellular matrix (ECM) that forms a barrier for T cells. Comparing five preclinical mouse tumor models with heterogeneous tumor microenvironments, we aimed to relate the rate of tumor stiffening with the remodeling of ECM architecture and to determine how these features affect intratumoral T cell migration. An ECM-targeted strategy, based on the inhibition of lysyl oxidase, was used. In vivo stiffness measurements were found to be strongly correlated with tumor growth and ECM crosslinking but negatively correlated with T cell migration. Interfering with collagen stabilization reduces ECM content and tumor stiffness leading to improved T cell migration and increased efficacy of anti-PD-1 blockade. This study highlights the rationale of mechanical characterizations in solid tumors to understand resistance to immunotherapy and of combining treatment strategies targeting the ECM with anti-PD-1 therapy.

## Introduction

In the last decade, significant progress has been made in the development of T-cell-based immunotherapies (*Miller and Sadelain, 2015*). The two main T-cell-based immunotherapies are adoptive T cell therapy and immune checkpoint inhibitors. Monoclonal antibodies blocking the immune checkpoints cytotoxic T lymphocyte-associated antigen 4 (CTLA-4) and programmed death one receptor (PD-1) have quickly gone from proof of concepts to FDA-approved first- and second-line treatments for a significant number of tumors even in late stages (*Callahan et al., 2016*). However, an elevated percentage of patients with solid tumors fail to respond to these therapies. The mechanisms

underlying the poor response to immune checkpoint inhibitors are still uncertain; nevertheless, recent results suggest that T cell function and distribution in the tumor are affected by numerous immunosuppressive mechanisms (*Anderson et al., 2017*). It is well established that in progressing tumors T cells exhibit a particular phenotype unable to normally respond to tumor antigens. In addition, in a large proportion of tumors, T lymphocytes are excluded from the tumor cell regions in a so-called 'excluded-immune profile' (*Hegde et al., 2016*; *Herbst et al., 2014*; *Joyce and Fearon, 2015*). Ineffective T cell migration and penetration into the tumor mass might represent an important obstacle to T cell-based immunotherapies. As a support for this notion, various clinical studies have shown that tumors enriched in T cells are more susceptible to be controlled by PD-1 blockade. In contrast, tumors with so-called immune-excluded profiles, in which T cells are present within tumors but not in contact with malignant cells, are refractory to PD-1 blockade (*Herbst et al., 2014*; *Mariathasan et al., 2018*). Particularly, the fibrotic state of desmoplastic tumors can cause immunosuppression through multiple mechanisms (*Turley et al., 2015*). The hypothesis of physical resistance to T cell infiltration and migration-related to the heterogeneity and aberrant organization of the extracellular matrix (ECM) with respect to the tumor nests has emerged recently (*Jiang et al., 2017*; *Pickup et al., 2014*). By using dynamic imaging microscopy, we highlighted the detrimental impact of collagen fibrils architecture on the migratory behavior of T cells in fresh human tumor explants. Both a guiding strategy combined with a physical hindrance process has been shown to restrain T cells from contacting tumor cells, thus leading to the T cell excluded profile (*Salmon et al., 2012*; *Peranzoni et al., 2013*). Hence, a dense fibrotic stroma could raise physical obstacles to immune cell infiltration similar to the previously established stromal resistance to chemotherapeutics, antibodies, nanoparticles, or virus tumor penetration (*Netti et al., 2000*; *Stylianopoulos et al., 2018*). In addition, cellular components of tumor-associated fibrosis, particularly the cancer-associated fibroblasts (CAF), can have both direct and indirect effects on T cell infiltration and function (*Turley et al., 2015*). Accordingly, one important challenge in the field is to develop strategies targeting tumor fibrosis in order to reverse immune exclusion and to improve T cell-based immunotherapy. Recent studies have been undertaken with this objective. T cells engineered to express a chimeric antigen receptor together with heparanase, an ECM-degrading enzyme, show enhanced infiltration into xenografted tumors as well as anti-tumor efficacy (*Caruana et al., 2015*). Recently, a major role for the TGFβ signaling pathway in promoting T cell exclusion from tumor cells has been demonstrated. In breast and colorectal mouse tumor models, neutralizing antibodies against TGFβ were shown to reduce collagen I production, overcoming the T cell excluded profile and increasing the efficacy of anti-PD-L1 antibodies (*Mariathasan et al., 2018*; *Tauriello et al., 2018*). In cholangiocarcinoma, an immune mesenchymal subtype has been identified, which is associated with TGFβ signature and poor tumor-infiltrating cells (*Job et al., 2020*). Other axes including the CXCR4/CXCL12 in breast metastasis and the focal adhesion kinase in pancreatic ductal adenocarcinoma (PDAC) have also been associated with both desmoplasia and absence of cytotoxic T lymphocytes in tumors from mice (*Chen et al., 2019*). Consequently, the inhibition of these axes in preclinical mouse cancer models was shown to reduce fibrosis while significantly increasing T cell infiltration and improving response to checkpoint inhibitors (*Mariathasan et al., 2018*; *Tauriello et al., 2018*; *Incio et al., 2015*; *Jiang et al., 2016*). Clinical trials testing such combination are currently ongoing in advanced pancreatic cancer, mesothelioma, urothelial carcinoma, and other malignancies (NCT02546531, NCT02758587, NCT02734160, NCT04064190, and NCT02947165).

However, due to patient and tumor heterogeneity, there is no clear indication of how the T cell distribution in tumors is related to the fibrosis level and to the different architectures of ECM. Thus, there is an urgent need to assist in matching combination approaches to patient populations who could benefit from stromal modulation strategies to improve their response to immunotherapy. Companion matrix-derived biomarkers and imaging approaches should provide insights into the contribution of ECM remodeling in shaping the immune milieu of the tumor. Particularly, a critical determinant of fibrotic tumor progression – the tumor mechanics – has been poorly investigated through the prism of immune impact. An important feature of fibrotic tumors is their considerable higher stiffness compared to their neighboring healthy tissues, which are highly correlated with cancer progression and metastasis, particularly in breast, colorectal, liver, and pancreatic tumors (*Venkatesh et al., 2008*; *Samani et al., 2007*). The use of non-invasive imaging techniques such as shear wave elastography (SWE) and magnetic resonance elastography, designed to monitor stiffness of any given tissue, allows an accurate and non-invasive diagnostic and characterization of malignant

lesions in vivo with prognostic significance, for instance in breast cancer (*Evans et al., 2012*; *Song et al., 2018*; *Riegler et al., 2018*). Indeed, the extensive remodeling of the stromal components which increase tumor stiffness can mechanically activate intracellular signaling pathways that promote tumor progression and at the same time can dampen T cell functions including migration and infiltration into tumor islets (*Datar and Schalper, 2016*; *Humphrey et al., 2014*; *Krebs et al., 2017*; *Rice et al., 2017*). However, there is a lack of studies correlating the mechanical properties of tumors to their heterogeneous ECM architecture and T cell infiltration capacity. Here we aim at filling this gap through a comprehensive investigation of stiffness evolution in several preclinical mouse models of pancreatic, breast, and bile duct carcinomas, presenting different ECM organizations, coupled with dynamic imaging of fresh tumor slices to monitor T cell motility. In concert with these imaging biomarkers of both mechanical properties, ECM architecture, and T cell migration, we explored the consequences of altering the ECM by inhibition of the lysyl oxidase (LOX), a copper-dependent enzyme responsible for the crosslinking of collagen molecules into fibers that has been seen to be overexpressed in many metastatic tumors and responsible for malignant progression (*Cox et al., 2016*). We highlight that LOX inhibition has different mechanical modulating effects depending on the ECM architecture, with significant improvement in T cell mobility. Despite minor effects in primary tumor growth upon LOX inhibition or PD-1 blockade treatment alone, their combination increases effector CD8 T cell accumulation in tumors and significantly delays tumor progression in a pancreatic cancer model.

## Results

### Relationship between tumor structure and tumor mechanical properties in different preclinical carcinoma mouse models

One key aspect when testing immunotherapeutic agents is the use of relevant preclinical models that closely mimic the properties of human solid tumors. Human carcinomas derive from epithelial cells and therefore harbor a typical though heterogeneous structure with tumor cells forming compact islets or nests surrounded by the stroma, enriched in ECM proteins, fibroblasts, blood vessels, and immune cells. To unravel the relationship between tumor growth, ECM remodeling, stiffening, and immune infiltration, we characterized the tumor structure and the mechanical properties of five different preclinical models, recapitulating the structural heterogeneity of different carcinomas (*Supplementary file 1*): a subcutaneous model of cholangiocarcinoma (EGI-1), a subcutaneous (MET-1) and a transgenic model (MMTV-PyMT) of mouse breast carcinoma, and an orthotopic (mPDAC) and a subcutaneous (KPC) model of mouse PDAC. A multiscale evaluation of the mechanical properties of the tumors was performed. At the macroscale, we measured tumor stiffness during tumor growth using SWE, a non-invasive imaging technique that allows the quantification and mapping of tumor stiffness (*Figure 1A*, *Supplementary file 2*). The presence of very stiff regions, defined as areas with an elastic modulus > 40 kPa (*Marangon et al., 2017*), in the tumor was quantified together with the average stiffness of the tumor (*Figure 1B*, *Supplementary file 2*). At the micron-scale, we evaluated tumor organization and fibrosis using hematoxylin–eosin–Safran (HES) (*Figure 1C*) and Sirius Red staining (*Supplementary file 2*). Sirius Red is a highly specific stain for collagen fibers that combined with polarizing microscopy allows differentiating thin collagen fibrils from thick and densely packed collagen fibers (*Rittié, 2017*). Under polarized light, thin fibers show a greenish-yellow birefringence, whilst thicker and densely packed fibers give an orange-red birefringence. By separating these two colors, it is possible to quantify the amount of thick and densely packed fibers present in the tumor (*Supplementary file 2*). The fibrillar collagen network was determined using second-harmonic generation (SHG) imaging, which allows to analyze the architecture and density of fibrillar collagen without having to use detection antibodies (*Figure 1D*, *Supplementary file 2*).

In the EGI-1 cholangiocarcinoma model, tumor stiffening and tumor growth have a strong positive correlation (*Figure 1A*). The stiffness distribution is highly heterogeneous, presenting 20% of stiff regions (>40 kPa) on average that goes up to 50–60% in tumors with higher volume. In terms of architecture, the tumor and its extensive stroma compartment, occupying around 20% of the tumor, are well separated. This is a typical trait of desmoplastic tumors and the model accurately reproduces the architecture of human cholangiocarcinoma. Its collagen network is characterized by long

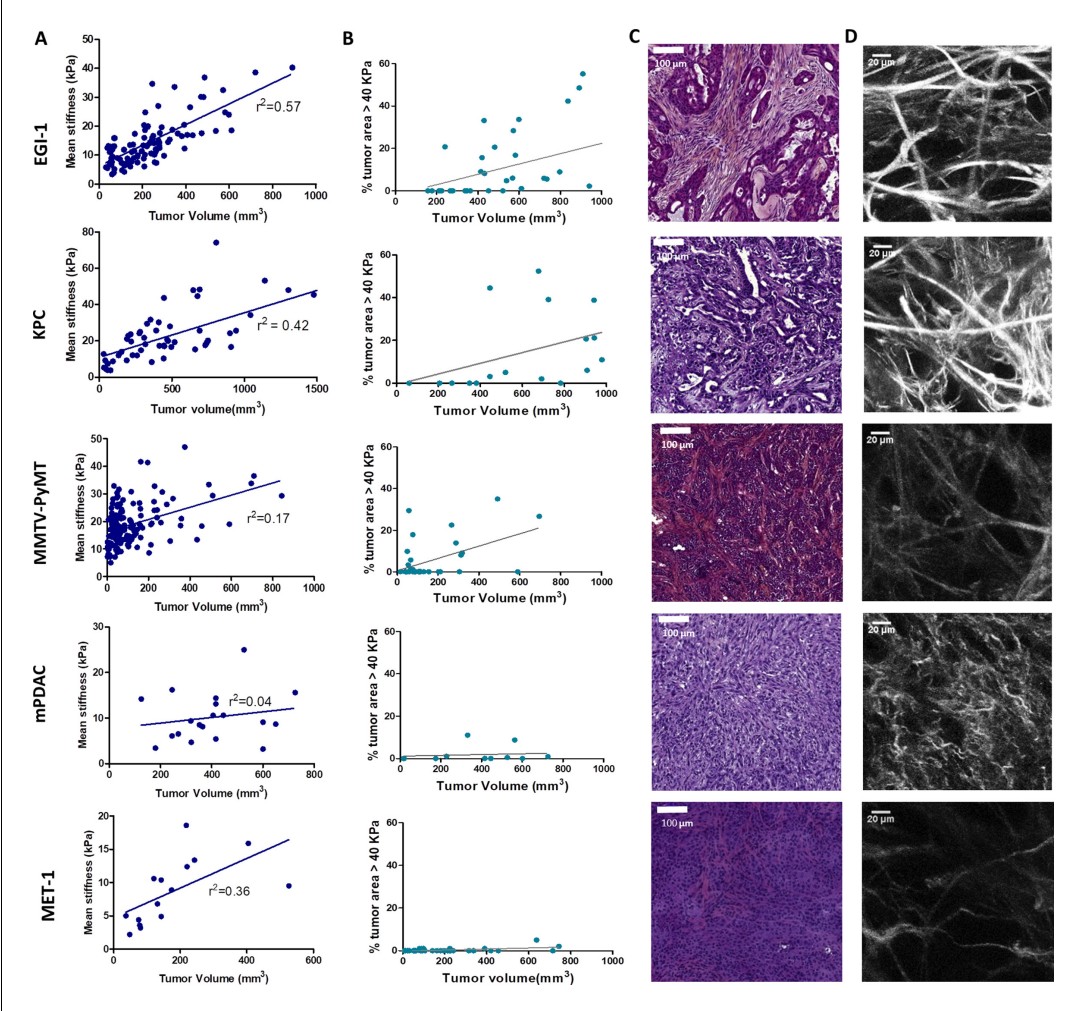

**Figure 1.** Macroscopic and microscopic characterization of EGI-1, KPC, MMTV-PyMT, mPDAC, and MET-1 tumor models. (A) Tumor volume and tumor mean stiffness relationship. Tumor volume was measured using a caliper or ultrasound, whilst tumor mean stiffness was measured using SWE. (B) The percentage of stiff regions in relation to tumor volume. The presence of very stiff regions with an elastic modulus > 40 kPa was quantified using data from the stiffness map extracted from the SWE images. (C) Comparison of the histological diversity within the tumor models. Representative images of each tumor model (scale bar = 100 µm). (D) SHG images of the collagen network in each of the models at the end point of the experiment. Scale bar = 20 µm. EGI-1 (n = 30 mice from three independent experiments); KPC (n = 34 mice from three independent experiments); MMTV-PyMT (n = 5 mice, 10 tumors per mouse from two independent experiments); mPDAC (n = 7 mice from two independent experiments); and MET-1 (n = 12 mice from two independent experiments).

The online version of this article includes the following source data for figure 1:

**Source data 1.** Source data file for *Figure 1*.

(85.9 ± 42.0 µm) and thick (7.4 ± 2.2 µm) collagen fibers that are densely packed (6% of the tumor) (*Figure 1D*, *Supplementary file 2*).

The mouse PDAC KPC model (*Hingorani et al., 2005*) exhibits similar features to that of EGI-1. A high positive correlation between tumor stiffness and volume, presenting over 20% of stiff regions at high tumor volumes, was observed (*Figure 1A*). KPC tumors present typical segregation of ECM and tumor nests, with a high proportion of stroma (~20%). However, the extension of the stromal areas is lower than that of EGI-1 with higher intercalation with tumor islets. Its collagen network is characterized by shorter (67.1 ± 35 µm) and thinner (4.4 ± 0.9 µm) collagen fibers than that of EGI-1 (*Figure 1D*, *Supplementary file 2*).

The spontaneous orthotopic murine breast cancer model MMTV-PyMT, although slower in its growth as compared to subcutaneous tumors, also stiffens during tumor progression (*Figure 1A*). However, there is a lower density of stiff regions (<16%). Of note, the fact that it is an orthotopic

tumor in a genetic model that develops about 10 tumors limits the maximal tumor volume reached for this analysis. Hence, we cannot compare this model with the other models at high tumor volumes for ethical reasons. This spontaneous tumor model also presents a tumor islets-stroma structure, but with a lower amount of stroma (~13%) as compared to EGI-1 and KPC models. The stroma is more dispersed and intercalated with the tumor compartment. Collagen fibers are characterized for being thin (3.6 ± 1.1 µm) and long (83 ± 41 µm), forming densely packed regions taking up to 4.8% of the tumor.

The orthotopic murine PDAC (mPDAC) model has a very different profile compared to the other models. The stroma takes up ~40% of the tumor (*Figure 1C*, *Supplementary file 2*) but without clear spatial segregation of stromal and tumor compartments. The mPDAC collagen network is made up of thin (3.5 ± 0.8 µm) and dispersed collagen fibers, which accounts for the lower presence of densely packed collagen regions (2.7% of the tumor). The mean tumor stiffness is lower than that of the other models, partly explained by the limits of the maximal tumor volume reached in this orthotopic model (for ethical reasons) and partly by the collagen architecture. Tumor stiffness also increases with tumor volume (*Figure 1B*) in line with previous studies performed in this model (*Gilles et al., 2016*). Unlike the above-mentioned models that exhibit a high correlation between stiffness and tumor growth, the mouse breast carcinoma MET-1 tumor model is characterized by low tumor stiffness (*Figure 1A*), a limited stroma (~6%), the lack of tumor-islet/stroma organization, and the presence of thin (3.2 ± 0.9 µm) and dispersed collagen fibers (*Figure 1D*, *Supplementary file 2*).

In human breast cancer, a significant correlation between tumor stiffness and tumor size was demonstrated (*Evans et al., 2012*; *Song et al., 2018*). Here, our analysis enabled us to confirm such correlation in different mouse tumor models covering three types of carcinomas. In addition, we show a correlation between tumor stiffness measured non-invasively with collagen accumulation associated with a segregated architecture of thick and densely packed collagen fibers (Sirius red positive) surrounding tumor nests. In contrast, tumors with an entangled and thin mesh of collagen fibers present lower rigidity despite overall high collagen content. Particularly the appearance of stiff regions > 40 kPa is seen as a physical biomarker of intratumor heterogeneity and ECM segregation. This analysis maps out potentially relevant preclinical tumor models, which might reproduce the diverse fibrotic evolutions of human breast, pancreatic, and bile duct tumors and their architecture heterogeneity.

## LOX modulates tumor stiffness and the ECM organization

The panel of tumor stroma structures reported above allows us to investigate the direct effects of ECM modulating agents in situations mimicking the heterogeneity observed in human carcinoma. Thus, we sought to determine whether beta-aminopropionitrile (BAPN), an inhibitor for LOX enzymatic activity, could modulate tumors' mechanical properties in concert with the stroma architecture (*Leventa et al., 2009*). For these experiments, BAPN was administered in the drinking water of mice upon tumor cell implantation and until their sacrifice for most models, except for the MMTV-PyMT model that was treated approximately at the time that tumors start to spontaneously develop. LOX stabilizes collagen fibers by enzymatic reactions that culminate in the formation of trivalent mature crosslinks including pyridinoline (PYD) and deoxypyridinoline (DPD) (*Yamauchi et al., 2018*). PYD and DPD have an intrinsic fluorescence (*Richards-Kortum and Sevick-Muraca, 1996*), which can be measured by two-photon microscopy (Ex 720 nm, Em 400 nm) on tissue sections (*Marturano et al., 2014*). Here, we adopted a similar strategy and assessed these two LOX-generated crosslinks in tumor slices from control and BAPN-treated KPC-bearing mice. Our data indicate that the average fluorescence signals of PYD and DPD, measured in SHG-positive regions, were significantly decreased in BAPN as compared to control conditions (*Figure 2—figure supplement 1*). These results suggest that BAPN specifically blocks LOX enzymatic activity. We then examined the effect of LOX inhibition on tumor stiffness (*Figure 2A,C*) and on the presence of stiff regions (*Figure 2B*). Results show that all models, except for MET-1, undergo a reduction in mean stiffness when LOX is inhibited. EGI-1 and KPC models both show the most striking differences (*Figure 2A, C*). Changes are mainly perceived at late stages of tumor developments since, in these models, tumor stiffness is positively correlated with tumor growth. In the MMTV-PyMT model, however, significant differences were noted throughout the development of the tumor. For the mPDAC model, tumor stiffness was only evaluated at the end of the BAPN treatment. A significant decrease in mean tumor stiffness is seen in BAPN-treated mPDAC tumors.

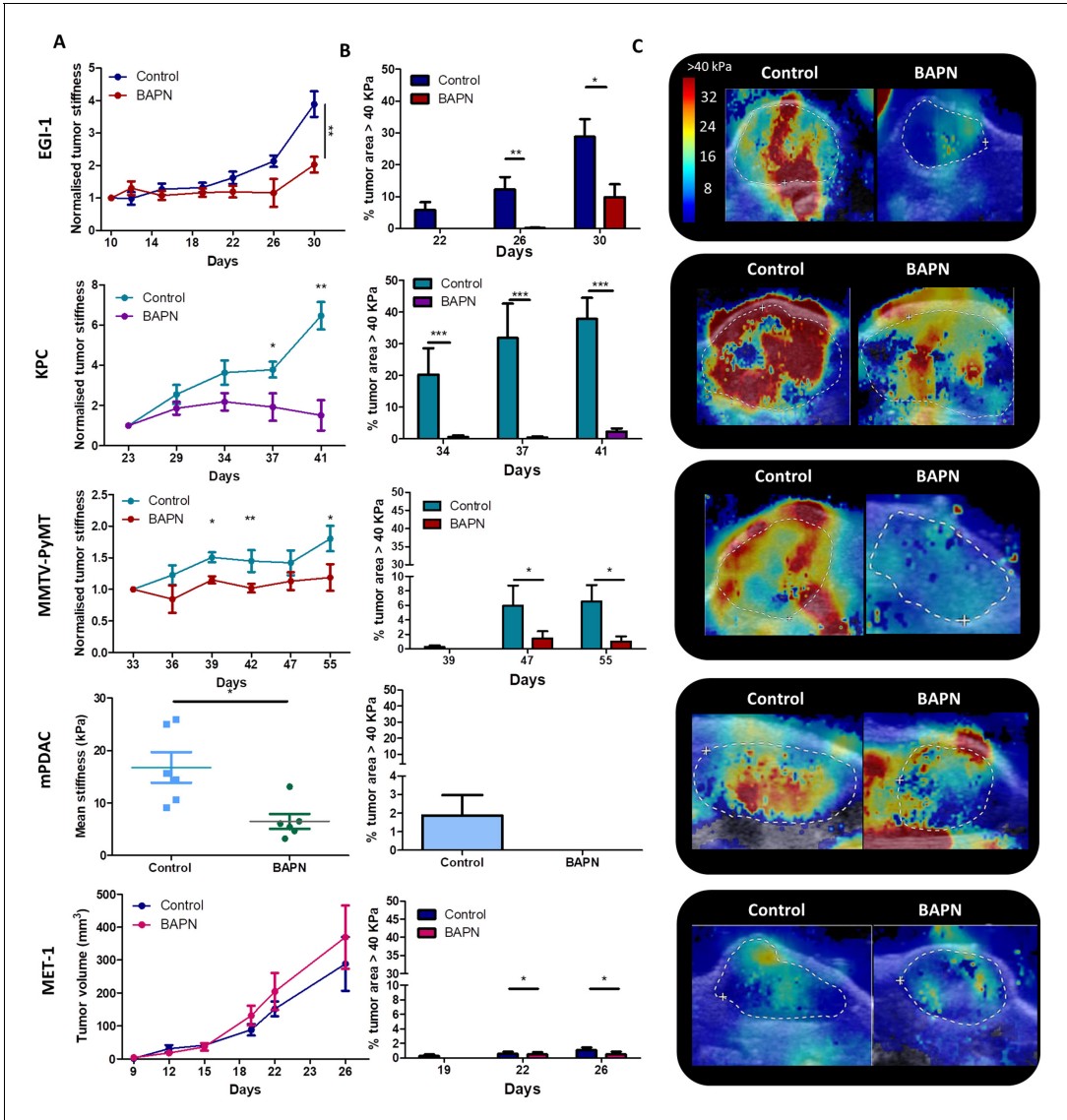

**Figure 2.** Effect of LOX inhibition on tumor stiffness and tumor stroma organization in EGI-1, KPC, MMTV-PyMT, and mPDAC tumor models. (**A**) Tumor stiffness measured by SWE in control and BAPN-treated (LOX inhibitor) tumors in relation to tumor volume for EGI-1, MMTV-PyMT, and KPC tumor models. For the mPDAC tumor model, stiffness was only measured at the endpoint of the experiment (*p-value<0.05, **p-value<0.01, ***p-value<0.001 Student's t-test). (**B**) Percentage of tumor area with stiffness > 40 kPa at three different time points in the late stages of tumor development. (**C**) Illustrative examples of SWE images of each on the tumors at the last time point of the experiment (EGI-1 – day 30, KPC – day 41, MMTV-PyMT – day 55, mPDAC – day 21). EGI-1 (n = 30 mice/group from three independent experiments); KPC (n = 34 mice/group from three independent experiments); MMTV-PyMT (n = 5 mice/group, 10 tumors per mouse from two independent experiments); mPDAC (n = 7 mice/group from two independent experiments), and MET-1 (n = 12 mice/group from two independent experiments).

The online version of this article includes the following source data and figure supplement(s) for figure 2:

**Source data 1.** Source data file for *Figure 2*.

**Figure supplement 1.** Two-photon imaging analysis of collagen crosslink density from control and BAPN-treated conditions.

**Figure supplement 1—source data 1.** Source data file for *Figure 2—figure supplement 1*.

**Figure supplement 2.** Effect of LOX inhibition in EGI-1 tumor model.

**Figure supplement 2—source data 1.** Source data file for *Figure 2—figure supplement 2*.

**Figure supplement 3.** Effect of LOX inhibition in KPC tumor model.

**Figure supplement 3—source data 1.** Source data for *Figure 2—figure supplement 3*..

**Figure supplement 4.** Effect of LOX inhibition in MMTV-PyMT tumor model.

**Figure supplement 4—source data 1.** Source data file for *Figure 2—figure supplement 4*.

**Figure supplement 5.** Effect of LOX inhibition in mPDAC tumor model.

*Figure 2 continued*

**Figure supplement 5—source data 1.** Source data file for *Figure 2—figure supplement 5*.
**Figure supplement 6.** Effect of LOX inhibition in MET-1 tumor model.
**Figure supplement 6—source data 1.** Source data file for *Figure 2—figure supplement 6*.

Notably, BAPN treatment did not affect tumor growth in most models (*Figure 2—figure supplements 2A–6A*), except for mPDAC (*Figure 2—figure supplement 5A*). To verify that the variation in tumor stiffness was not due to a difference in tumor volume, the mean tumor stiffness of control and treated mice were compared at different tumor volumes (*Figure 2—figure supplements 2C–6C*). In both KPC and EGI-1 tumors, a clear difference in mean tumor stiffness can be seen in tumors with a volume > 400 mm$^3$ (*Figure 2—figure supplements 2C–3C*).

We also explored whether the presence and proportion of stiff regions was reduced when LOX was inhibited (*Figure 2B*). The percentage of control tumor area with a mean stiffness > 40 kPa increased with time (and with tumor volume) within the non-treated tumor indicating that there is not only an increase of overall mean stiffness but also an increase of the heterogeneity of stiff regions. However, this percentage was significantly reduced in BAPN-treated tumors with marked differences observed in the KPC model and to a lesser extent in EGI-1 and MMTV-PyMT models. In mPDAC, BAPN-treated tumors did not display stiff regions (*Figure 2—figure supplement 5B–D*). The only model that does not respond to LOX inhibition by stiffness reduction is MET-1 (*Figure 2—figure supplement 6B–C*), in line with our previous data showing an absence of tumor stiffening during tumor growth. Overall, our results clearly illustrate: (1) the heterogeneity of tumor response to an ECM-targeting agent, (2) the potential of non-invasive SWE elastography to measure a macroscopic physical marker – stiffness – that predicts this response.

Given the effects of LOX inhibition at a macro scale, we decided to delve into the changes induced at the level of the collagen fiber network through an in-depth quantitative evaluation of collagen fiber width (*Figure 3A*), orientation (*Figure 3B,C*), curvature (*Figure 3D*), and the presence of regions with thick and densely packed fibers (*Figure 3E*).

A significant reduction of collagen fiber width distribution was observed in EGI-1, KPC, and mPDAC models, whilst fibers in the MMTV-PyMT tumor model did not display a significant change in their width (*Figure 3A*). The most substantial difference was seen in the EGI-1 model, where collagen fiber width was decreased by 9.4% on average (6.7 μm versus 7.4 μm). Changes in mPDAC and KPC were less pronounced, with a reduction of 4.4% and 5%, respectively (*Supplementary file 3*). The inhibition of LOX did not affect collagen fiber length in any of the models (*Supplementary file 3*).

We next assessed the orientation and linearization of collagen fibers in control and BAPN-treated animals. In general, the collagen fibers in normal tissues are typically curly and anisotropic in contrast to the situation observed in progressing tumors in which many of the fibers progressively thicken and linearize. Collagen fiber orientation was described as the coefficient of variation (CV) of the angle for all fibers, the smaller the CV is, the more aligned the fibers are. Fibers in non-treated tumors remained mainly oriented in one dominant direction (*Figure 3B, C, and F*) with a CV from 1.85 to 0.5 consistent with previous findings (*Li et al., 2019*). LOX inhibition tends to disrupt the alignment of collagen fibers, meaning that they were more dispersed and oriented in different directions with increased CV as compared to control conditions. The most significant effects were seen in KPC and mPDAC models.

Collagen fibers in tumors are characterized as being linear and reticulated due to the high level of deposition and posttranslational crosslinking. This physical restructuration of collagen progressively stiffness the ECM (*Egeblad et al., 2010*). The level of collagen linearization was quantified by measuring the curvature ratio of the fibers. The curvature ratio of control tumors for all models was close to one meaning that the collagen fibers were fully linearized. In contrast, LOX inhibition severely affected fiber curvature, as there was a reversion of the fibers linearization resulting in less linear and wavier fibers in all models (*Figure 3D,F*).

Finally, LOX inhibition also significantly decreased the surface covered by thick and densely packed collagen fibers in EGI-1, KPC, and MMTV-PyMT model (*Figure 3E*, *Figure 3—figure*

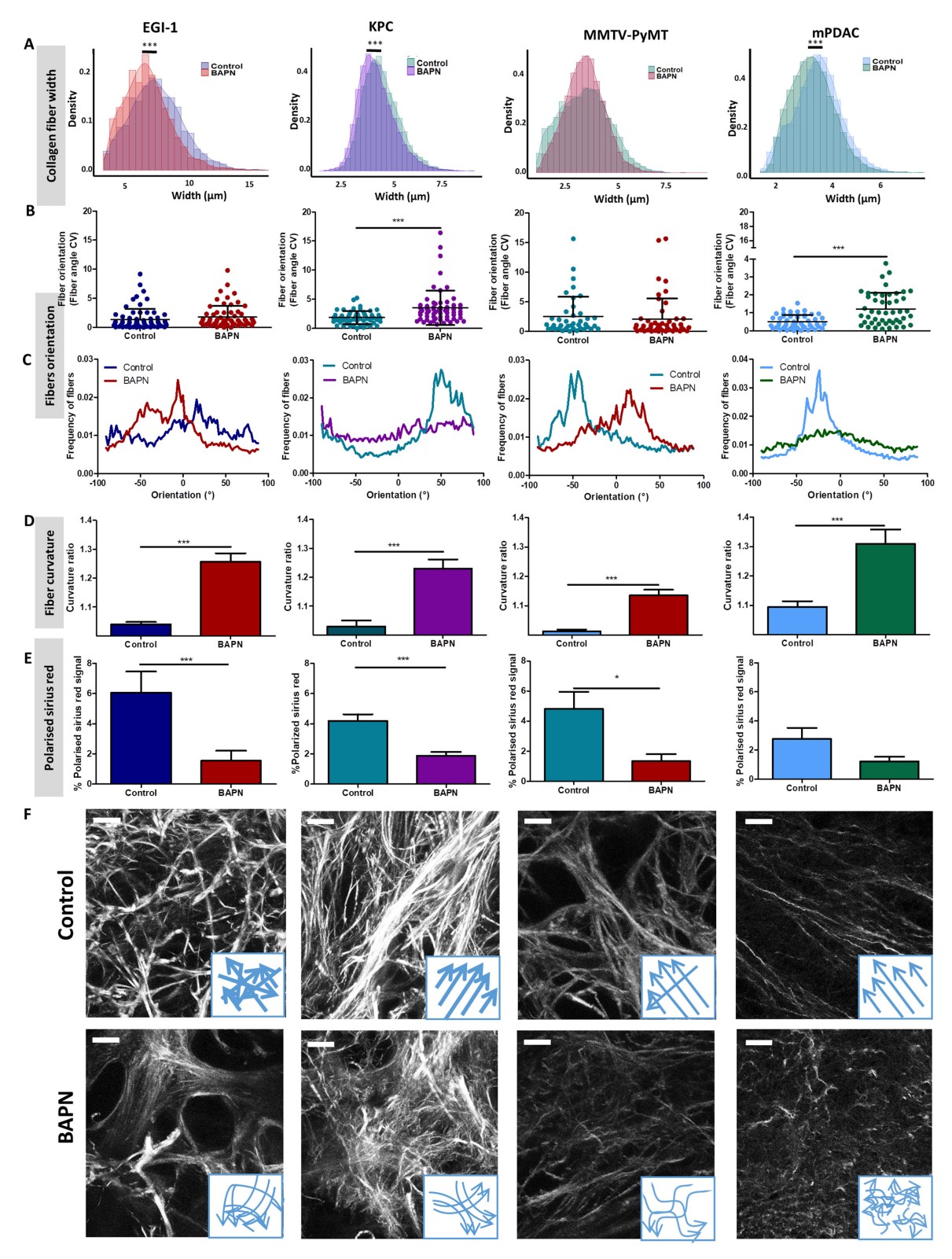

**Figure 3.** Effect of LOX inhibition on ECM architecture in EGI-1, KPC, MMTV-PyMT, and mPDAC tumor models. (**A**) Collagen fiber width distribution measured from SHG images (***p-value<0.001, Student's t-test, 50–70 images/tumor, n = 3 mice/group from three independent experiments). (**B**) Fiber orientation defined by the coefficient of variation (CV) (***p-value<0.001, Student's t-test, 40–50 images/tumor, n = 3 mice/group from three independent experiments). (**C**) Representative example of fiber orientation distribution. (**D**) Collagen fiber curvature defined by the curvature ratio

*Figure 3 continued on next page*

*Figure 3 continued*

(***p-value<0.001, Student's t-test, 40–50 images/tumor, n = 3 mice/group from three independent experiments). (**E**) Percentage of red-orange birefringent fibers combining Red Sirius staining and polarized microscopy, orange-red fibers correspond to thick and packed regions (*p-value<0.05, ***p-value<0.001, Student's t-test, 20–30 images/tumor, n = 3 mice/group from three independent experiments). (**F**) SHG images of collagen networks in EGI-1, MMTV-PyMT, mPDAC, and KPC control and BAPN-treated tumors. Illustrative scheme indicating the changes in width, orientation and curvature of collagen fibers induced by BAPN treatment. Scale bar = 50 µm. Images are representative of three experiments.

The online version of this article includes the following source data and figure supplement(s) for figure 3:

**Source data 1.** Source data file for *Figure 3*.

**Figure supplement 1.** Combination of Sirius Red staining and polarized microscopy.

*supplement 1*), correlating with the significant decrease in average tumor stiffness and in the proportion of stiff areas measured in vivo.

In vivo stiffness imaging and ex vivo characterization of the ECM structure were completed by ex vivo mechanical evaluation of isolated tissue samples with atomic force microscopy (AFM) nanomechanical measurements and plate shear rheometry at the tissue level in the KPC model. *Figure 4* shows the mean values of storage modulus (G') of control and BAPN-treated tumors. It can be noted that tumors treated with BAPN were softer than control ones, with mean stiffness of 1.80 ± 0.51 kPa compared to 4.15 ± 1.92 kPa for the control. These results are in line with SWE measurements in vivo confirming a significant reduction in the mean stiffness of KPC tumors treated with a LOX inhibitor. In contrast to bulk rheometry, AFM reveals the spatial heterogeneity of Young's moduli at the sub-cellular level measured on six controls (*Figure 4—figure supplement 1*) and six BAPN-treated tumors (*Figure 4—figure supplement 2*). For each sample, high heterogeneity in tumor stiffness can be observed, as previously seen in other types of solid tumors like breast cancer

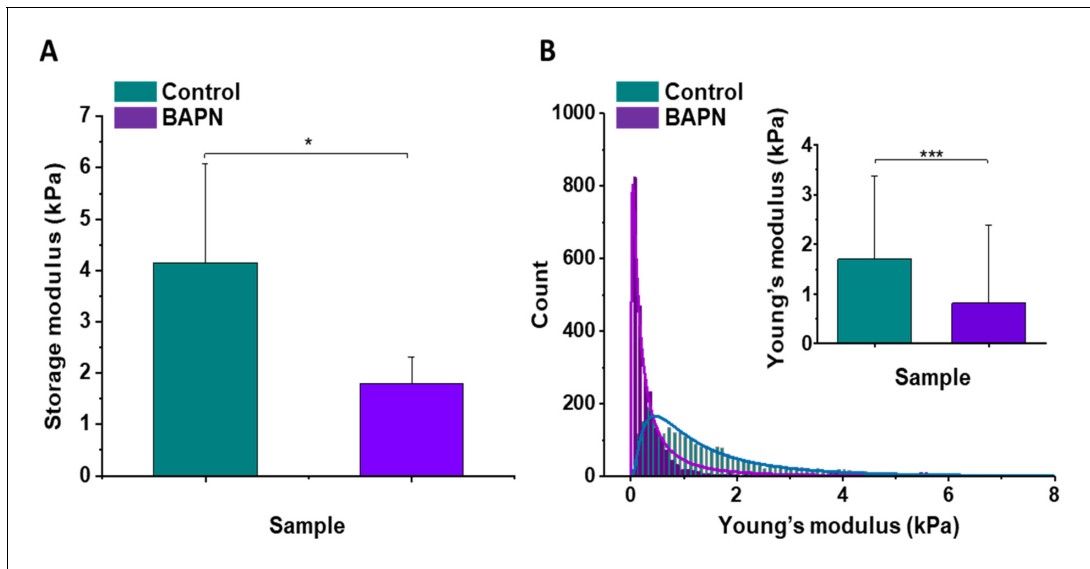

**Figure 4.** Mechanical properties of the mouse pancreatic ductal adenocarcinoma KPC tumor model and effects of LOX inhibition. (**A**) Rheological properties of tumor samples measured using a plate shear rheometer. Mean storage modulus (G') for all control and BAPN-treated samples ± SD are presented (for 5% sample compression). (**B**) The Young's modulus values distributions obtained for all control and treated tumor samples using the AFM indentation technique. Inset in (**B**) shows tissues' Young's modulus mean values ± standard deviation to highlight the difference between control and BAPN tissues. Statistical significance was determined using a two-tailed Student's t-test for overall values (***p-value<0.001, Student's t-test, n = 6 tumors/group from three AFM experiments; *p-value<0.05, Student's t-test, n = 5 tumors/group from three rheometer experiments).

The online version of this article includes the following source data and figure supplement(s) for figure 4:

**Source data 1.** Source data file for *Figure 4*.

**Figure supplement 1.** The distributions of Young's moduli obtained for the control mouse pancreatic ductal adenocarcinoma.

**Figure supplement 1—source data 1.** Source data file for *Figure 4—figure supplement 1*.

**Figure supplement 2.** The distributions of Young's moduli obtained for the BAPN-treated mouse pancreatic ductal adenocarcinoma.

**Figure supplement 2—source data 1.** Source data file for *Figure 4—figure supplement 2*.

(*Plodinec et al., 2012*). AFM measurements indicated that treatment of mice with BAPN leads to a narrower Young's modulus distributions shifted to lower values of elastic modulus in comparison to the control samples (mean Young's modulus of 0.82 ± 1.58 kPa versus 1.70 ± 1.66 kPa, respectively) (*Figure 4B*). This confirms that the high heterogeneity in local tissue mechanical properties of KPC tumor can be reduced upon treatment with a LOX inhibitor. Both local and global measurements confirm the normalization of tumor tissue mechanical properties mediated by LOX inhibition with a drastic reduction in the linearized tightly packed collagen fibers that contribute to tumor stiffness heterogeneity and global enhancement in non-treated KPC tumors.

Overall, our results demonstrate the rationale of targeting LOX enzymatic activity for normalizing tumor mechanical properties and ECM structure (mostly collagen fibers compaction, segregation, and linearity) in tumors exhibiting high tumor stiffness together with mechanical and structural heterogeneity.

## LOX inhibition increased intratumoral T cell migration and infiltration

Previous studies performed in our group have proved that the density and orientation of the ECM can have an important impact on T cell behavior and their displacement in fresh human lung and ovarian tumor slices (*Salmon et al., 2012*; *Bougherara et al., 2015*). Motile T cells were mainly found in loose ECM stromal regions, whereas fibrotic areas were devoid of lymphocytes. Based on this, we hypothesized that LOX-dependent tumor stiffness could consequently affect T cell migration in tumors and eventually predict the T cell behavior in the various ECM environments. To test this, we performed dynamic imaging of T cell migration on fresh tumor slices from mice treated or not with BAPN. The tumor slice assay that we have established preserves the original tissue microenvironment and permits monitoring with confocal microscopy the behavior of either ex vivo purified and plated T cells or endogenous T cells labeled with directly coupled fluorescent antibodies (*Peranzoni et al., 2018*). EGI-1 is a xenografted tumor model, with implantation of human carcinoma cells into immune-suppressed mice that lack resident T cells. Thus, in order to evaluate T cell migration in this model we isolated human peripheral blood T cells (PBT) and activated them in vitro. We then added the activated PBT onto fresh tumor slices and analyzed their migration using real-time confocal microscopy. As the MMTV-PyMT tumor model is poorly infiltrated in host T cells (*Guerin et al., 2019*), we investigated the migration of exogenously purified murine-activated PBTs in the same manner as for the human EGI-1 model. In both mPDAC and KPC mice tumor models, resident tumor-infiltrating T lymphocytes were monitored after staining with directly coupled anti-CD8 antibodies (*Peranzoni et al., 2018*). The three parameters analyzed to assess T cell migration were cell migration speed (mean cell migration speed over 20 min), cell displacement (displacement vector between starting and final position), and straightness (ratio of cell displacement to the total length of the trajectory) of the migration trajectory (*Figure 5* and *Supplementary file 4*).

In control untreated tumors, T cells migrated slowly with average velocities which were relatively homogeneous in the different models except in the mPDAC model (*Video 1*). Average velocities ranged from 2.3 µm/min in the EGI-1 model to 4.1 µm/min in the MMTV-PyMT model. These values are in line with a number of studies including ours showing a poor migration of T cells in tumors as compared to other organs (e.g. in lymph nodes) where T cells actively migrate (*Peranzoni et al., 2018*). For example, the mean velocity of CD8 T cells in human lung tumors approaches 3 µm/min (*Peranzoni et al., 2018*). Analysis of T cell track straightness gives indices close to 0.4 consistent with previous reports on T cell displacements in tumors.

The mPDAC model differs from the others since T cells were almost static during the 20 min recording (average speed of 1 µm/min and straightness of 0.1).

In every tested model, LOX inhibition resulted in an overall increase in T cell migration as compared to control conditions (*Figure 5*, *Supplementary file 4* and *Video 2*). However, different parameters were altered in each model depending on the nature of the T cells that were monitored. In EGI-1 BAPN-treated tumors, activated PBT cells displayed longer displacement lengths compared to untreated tumors. This was also true for MMTV-PyMT tumors. Since activated PBT are not specific to the tumor, the effects observed are due to LOX inhibition and not due to T cells engaging in stable conjugates with cancer cells through antigen recognition. In EGI-1 tumors, the trajectory straightness of activated PBTs was also significantly increased upon LOX inhibition.

Effects of BAPN on the dynamics of endogenous T cells infiltrated into KPC and mPDAC tumors were then evaluated (*Figure 5* and *Supplementary file 4*). We found that in both models, LOX

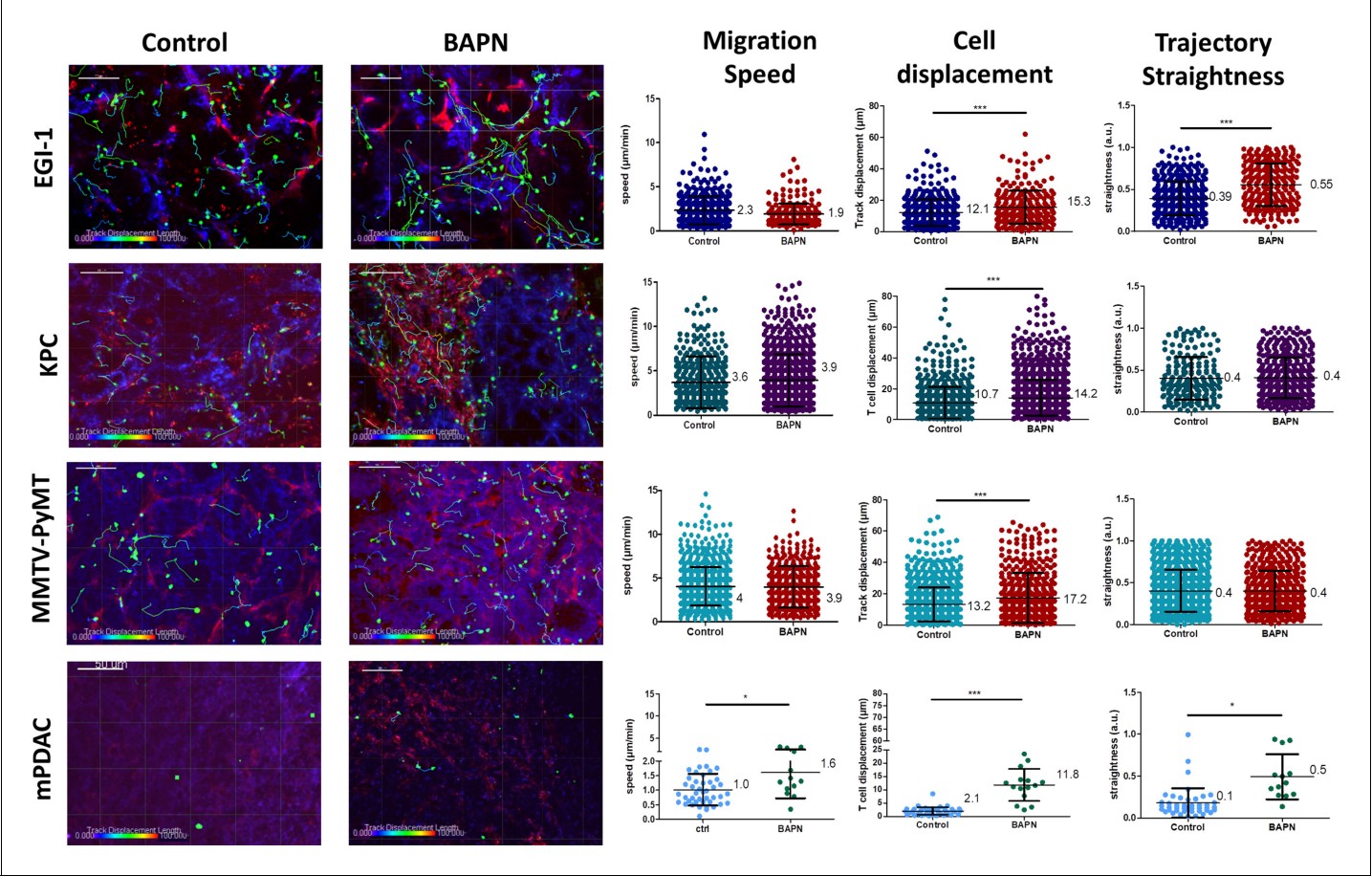

**Figure 5.** Impact of LOX inhibition on T cell migration in EGI-1, MMTV-PyMT, mPDAC, and KPC tumor models. Migration of activated PBT plated onto fresh tumor slices was analyzed in EGI-1 and MMTV-PyMT tumor model, whilst resident tumor-infiltrating T lymphocytes were analyzed in mPDAC and KPC tumor model. Illustrative images of T cell migration tracks in EGI-1, MMTV-PyMT, mPDAC, and KPC tumor models. Tumor stroma (fibronectin) in red, tumor cells (EpCAM in EGI-1, MMTV-PyMT, and KPC tumor models, CD44 in mPDAC tumor models), in blue and T cells (CD8 in mPDAC and KPC, Calcein in MMTV-PyMT, and EGI-1 tumor models) in green. Tracks are color-coded to illustrate track displacement. Scale bar = 100 µm. T cell migration speed, T cell displacement, and trajectory straightness in all tumor models. ***p-value>0.001, p-value>0.05, Student's t-test. Results are shown as mean ± SD.

The online version of this article includes the following source data and figure supplement(s) for figure 5:

**Source data 1.** Source data file for *Figure 5*.

**Figure supplement 1.** BAPN treatment induces an increased infiltration of effector CD8+T cells in KPC tumors.

**Figure supplement 1—source data 1.** Source data file for *Figure 5—figure supplement 1*.

inhibition leads to an increase in the displacement of T cells with a fivefold increase in the mPDAC model. In terms of cell speed, enhancements were observed only in the mPDAC model. Likewise, the trajectory straightness was increased in mPDAC models, but not in KPC tumors. In addition, we also evidenced an increased infiltration of resident T cells in KPC tumors when treated with BAPN. In control conditions, an identical number of T cells (around 100 per mm$^2$) were found in the stroma and tumor cell regions. By comparison, BAPN treatment resulted in a threefold to fourfold increase of CD8+ T cells in both the stroma and tumor islets (*Figure 5—figure supplement 1*).

Results reported in *Figure 5* have been obtained with data pooled from all mice either treated or not with BAPN. We decided to extend our analysis at the level of individual mouse and investigated the relationship between T cell motility (speed and displacement) and mean stiffness of control and BAPN-treated tumors. Our data indicate that T cell motility was inversely correlated to tumor stiffness (*Figure 6*) in line with *Figure 5*. However, in three out of the four models tested this correlation is different if one compares control and BAPN-treated tumors. In BAPN-treated tumors, there is a clear inverse linear correlation between T cell migration speed and mean tumor stiffness as

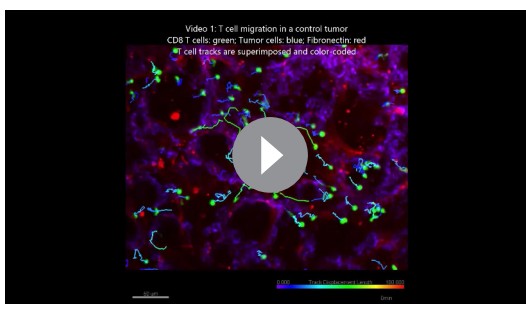

**Video 1.** T cell migration in the control EGI-1 tumor model. The slices stained with calcein for the T cells (green), anti-EpCAM (blue), anti-fibronectin (red) were imaged with a spinning disk confocal microscope. Tracks are color-coded according to T cell displacement length. Frame interval 30 s. The animations represent 3D reconstructions of sequential z series. A still image is shown in *Supplementary file 4*.
https://elifesciences.org/articles/58688#video1

evidenced by a steep slope. In comparison, such correlation is less pronounced in control stiff tumors. Similar results were observed when comparing T cell displacement and mean tumor stiffness. These results suggest that above a stiffness threshold that depends on the model, T cells are mostly arrested. In contrast, as tumor stiffness decreased by inhibition of LOX, T cell migration was restored. Thus, T cell motility is highly influenced by small variations in the stiffness of softer tumors as is the case when LOX's activity is inhibited.

Overall, these results suggest that the excessive accumulation and linearization of collagen in ECM limits T cell migration within several rigid tumors with desmoplastic evolution and that LOX-inhibiting BAPN treatment can both reverse tumor stiffening and improve T cell infiltration and migration to tumor cells. We also identify SWE tissue stiffness as a predictive physical marker of T cell motility and infiltration in desmoplastic tumors.

## LOX inhibition improves response to anti-PD-1 therapy

Even though the inhibition of LOX was followed by an increase in CD8 T cell number and migration, this finding was not accompanied by major effects on tumor growth in four of the five tumor models tested (*Figure 2—figure supplements 2–6*). In different settings, an increase in intratumoral T cell motility is not sufficient to reduce tumor growth if T cells are still impaired in their capacity to respond to tumor antigens (*Peranzoni et al., 2018*). Consequently, we decided to assess whether LOX inhibition could improve the response to immune checkpoint inhibitors. The KPC model was chosen for these experiments as tumors are stiff, respond to LOX inhibition in terms of ECM normalization, and are infiltrated with T cells. KPC tumor-bearing mice were treated with BAPN combined with anti-PD-1 antibodies. Mice were treated or not with BAPN from tumor cell injection up to their sacrifice and were treated with anti-PD-1 antibodies when the tumor volume was around 80–150 mm$^3$. At this point, the mice received four doses i.p. injection of anti-PD-1 or isotype control antibody at 4 days intervals (i.p. injection). Unlike BAPN, PD-1 blockade did not affect tumor stiffness (*Figure 7—figure supplement 1A,B*) nor collagen organization including fiber orientation and curvature (*Figure 7—figure supplement 1C–E*). Moreover, the combination therapy was similar to that of BAPN alone on all of these parameters. We then investigated the consequences of these treatments on tumor growth. As shown in *Figure 7A*, while treatment with single agents alone only shows non-significant reduction in tumor growth, the combination of BAPN with the checkpoint inhibitor significantly delays tumor progression. We then profiled the immune cell population in these tumors using flow cytometry and a gating strategy shown in *Figure 7—figure supplement 2*. We found that BAPN treatment alone significantly decreases the number of polymorphonuclear neutrophils (*Figure 7B*), but increases the presence of MHCII+ tumor-associated macrophages (TAMs)

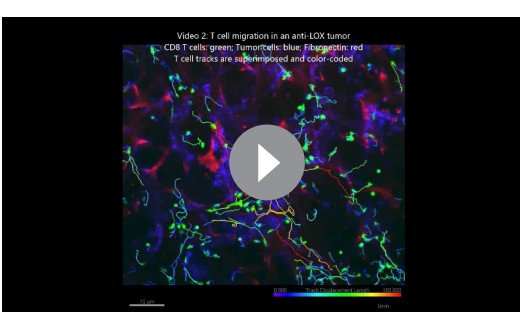

**Video 2.** T cell migration in the LOX inhibited EGI-1 tumor model. The slices stained with calcein for the T cells (green), anti-EpCAM (blue), and anti-fibronectin (red) were imaged with a spinning disk confocal microscope. Tracks are color-coded according to T cell displacement length. Frame interval 30 s. The animations represent 3D reconstructions of sequential z series. A still image is shown in *Supplementary file 4*.
https://elifesciences.org/articles/58688#video2

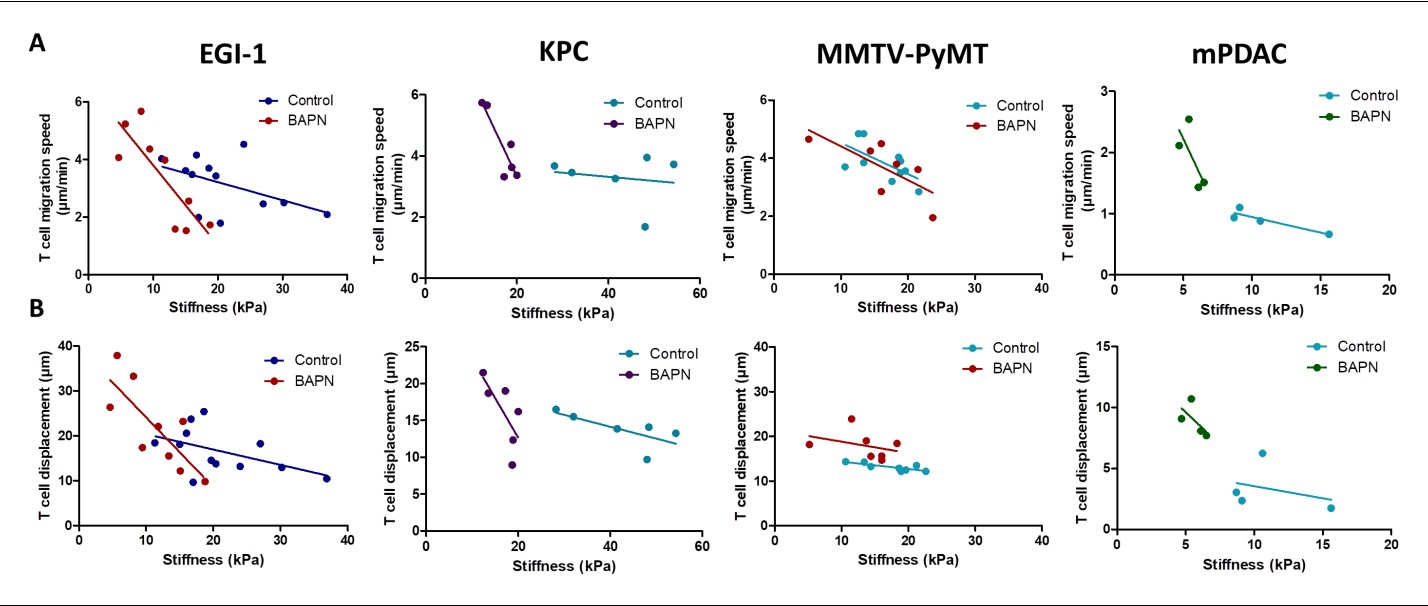

**Figure 6.** Correlation between mean tumor stiffness and T cell migration parameters in EGI-1, KPC, MMTV-PyMT, and mPDAC tumor models. (**A**) Correlation between mean tumor stiffness and mean T cell velocity. (**B**) Correlation between mean tumor stiffness and mean T cell displacement. Averaged T cell velocity and displacement were calculated from at least 50 individual cells. Each point represents values from an individual mouse. EGI-1 (n = 8 mice/group); KPC (n = 4 mice/group); MMTV-PyMT (n = 6–7 mice/group); and mPDAC (n = 4 mice/group) from two to three experiments. The online version of this article includes the following source data for figure 6:

**Source data 1.** Source data file for *Figure 6*.

(*Figure 7C*), while the combination therapy expanded the percentage of GrzmB CD8+ T cells (*Figure 7D*) and the ratio of CD8+ to Treg cells (*Figure 7E*). We also analyzed the amount of cytokines in supernatants of whole-tumor slices derived from these experiments. Results show that the combination therapy led to an increase in TNFα and RANTES, supporting further the increase of T cell activation and infiltration and activation in this condition (*Figure 7F–G*). In both BAPN and BAPN combined with anti-PD-1 conditions, we observed an increase of GrzmB+ levels compared to the control condition (*Figure 7H*). Of note, BAPN and anti-PD-1 alone also showed a similar tendency for an increase in TNFα and RANTES, which prompted us to study the consequences of these treatments on the intratumoral T cell motility. Consistent with the data in *Figure 5*, LOX inhibition leads to an increase in T cell displacement within the KPC tumor (*Figure 7—figure supplement 1F*). We also found that PD-1 blockade alone produced a similar increase in T cell motility, whereas the combination therapy showed the same effects as treatments alone (*Figure 7—figure supplement 1F*).

Overall, while ECM and stiffness normalization achieved through LOX inhibition increases T cell infiltration and migration, this strategy also improves the efficacy of anti-PD-1 blockade on tumor growth.

## Discussion

Despite the success of targeting the stromal compartment in tumors (*Mariathasan et al., 2018*; *Salmon et al., 2012*; *Caruana et al., 2015*; *Chen et al., 2019*; *Incio et al., 2015*; *Elahi-Gedwillo et al., 2019*), in particular tumor ECM, there are still a series of challenges that remain to be addressed. In the first part of this study, we tackle two of these challenges. One of them is finding ways to accurately assess the architecture of the stroma. In this paper, we propose a thorough analysis combining non-invasive imaging techniques for a macroscopic characterization of tumor stiffness with advanced microscopy techniques to elucidate the collagen network structure, one of the most important components of the tumor ECM. Previous studies had already explored the link between tumor stiffness and collagen architecture (*Venkatesh et al., 2008*; *Samani et al., 2007*;

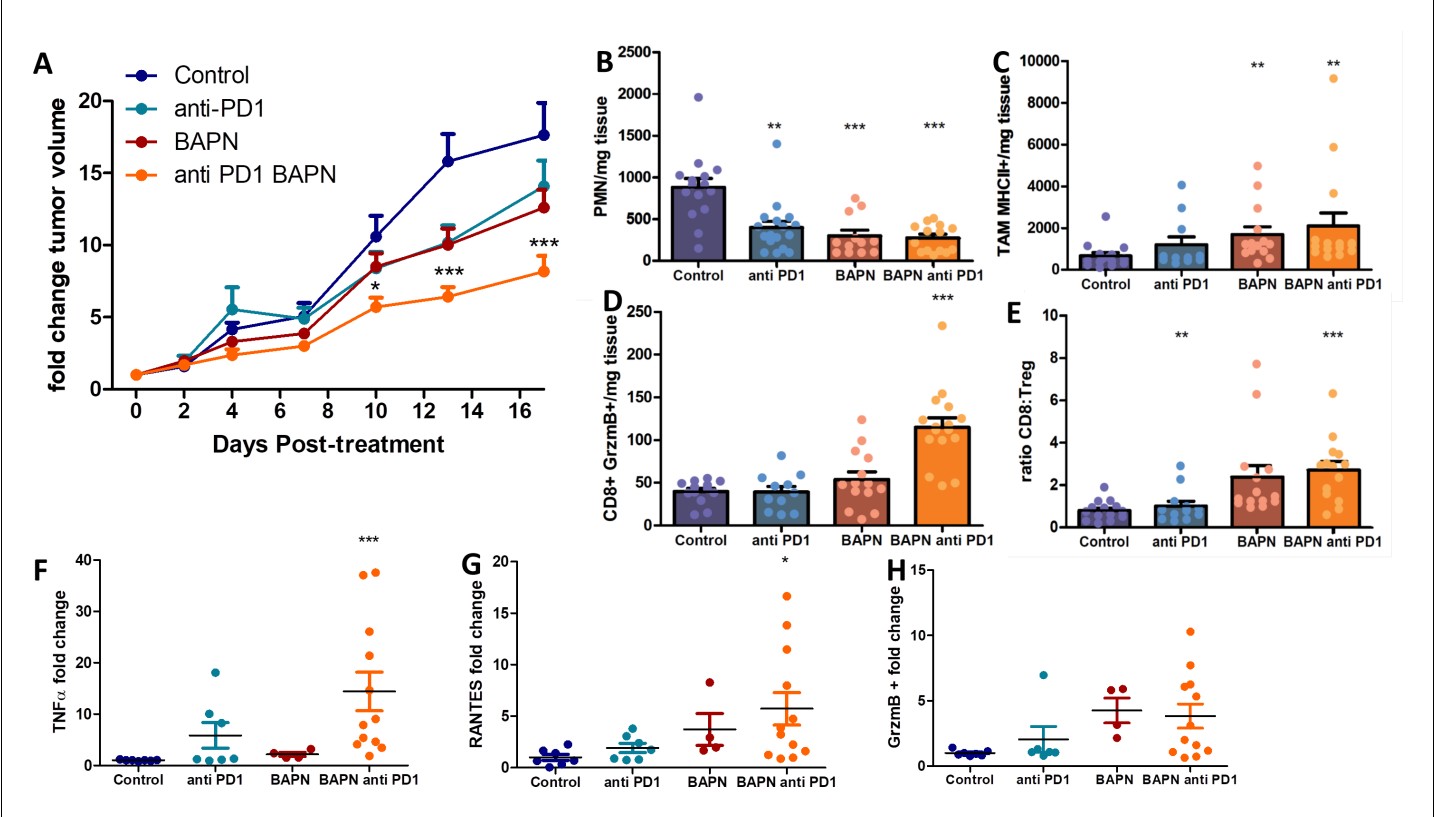

**Figure 7.** LOX inhibition increases the efficiency of anti-PD-1 therapy in KPC tumor model. (**A**) Tumor volume fold change after combination therapy of BAPN with anti-PD-1. n = 8 mice/group from two independent experiments. (**B**) CD11b+, Ly6G+, Ly6C+PMN/mg of tissue. (**C**) CD64+F4/80+MHCII +TAMs/mg of tissue. (**D**) CD8+ T cells/mg tissue. (**E**) CD8+ to FoxP3+Treg ratio. (**F–H**) Multiplex analysis of inflammatory chemokines (TNFα, RANTES, and GrzmB) produced by fresh KPC tumor slices kept in culture for 18 hr (*** p-value<0.001, * p-value<0.05, one-way ANOVA, Krustal Walis, n = 8 slices per condition from two independent experiments).

The online version of this article includes the following source data and figure supplement(s) for figure 7:

**Source data 1.** Source data file for *Figure 7*.
**Figure supplement 1.** Effect of LOX inhibition and anti-PD1 therapy on tumor stiffness, stroma organization and T cell motility in KPC tumors.
**Figure supplement 1—source data 1.** Source data file for Figure 7-figure supplement 1.
**Figure supplement 2.** FACS sequential gating strategies.

*Mieulet et al., 2021*). Here, we provide novel insights about this relationship through a multiscale analysis performed in several preclinical mouse models of pancreatic, breast, and bile duct carcinomas, presenting different ECM organizations and thus covering the heterogeneity found in cancer patients. This allowed us to extract mechanical and structural reference values and allowed us to later evaluate the effect of an ECM-targeted therapy. Such an approach could be translated to patients given the availability of SWE techniques that could be correlated to histological observations of the tumor microenvironment.

Accurately modeling tumor complexity and heterogeneity found in patients' tumors using a single preclinical model is a real challenge. In the first place, many preclinical models do not always reproduce the carcinoma structure found in their respective human tumors. Another important feature of human solid tumors, often absent in mouse tumor models, is their considerable higher stiffness compared to their neighboring healthy tissues, which is highly correlated with cancer progression and metastasis (*Evans et al., 2012*; *Song et al., 2018*; *Yoo et al., 2020*; *Evans et al., 2018*). These two features can often be found in human xenograft models that, on the other hand, are not suitable to study immune reactions. In this study, we have set up and characterized five different solid tumor models covering three types of carcinomas that differentiate in terms of tumor-stroma organization, tumor stiffness, collagen structure, and T cell infiltration. The xenotransplanted cholangiocarcinoma

tumor model (EGI-1) has an extensive stromal compartment well separated from tumor islets, which is a typical feature of desmoplastic tumors. Likewise, the KPC tumor model has also typical segregation of ECM and tumor nests and reproduces many of the key features observed in human PDAC including a stiff tumor and an exclusion of T cells from tumor islets. The orthotopic mPDAC model also contains very few T cells, although tumors are less stiff than the previous two models. The spontaneous MMTV-PyMT model is relevant with respect to human breast carcinoma as tumors developed in four stages (hyperplasia, adenoma/mammary intraepithelial neoplasia, and early and late carcinoma) (*Lin et al., 2003*). As a result, the architecture of MMTV-PyMT tumors resembles that of human breast cancers. On the other hand, the MET-1 model only contains thin stromal compartments and unlike the previous models, tumors are not very stiff, probably reflecting a specific subtype of human breast cancers. Once the tumor structure and mechanical properties of each model were characterized, an ECM-targeting therapy dependent on LOX inhibition with BAPN was tested. Previous studies performed by Levantal et al. in a spontaneous mouse model of breast carcinoma showed that LOX inhibition induces structural changes in the collagen network (*Leventant et al., 2009*). Here, we confirm and extend these results in other models and provide multiscale insights about different effects of LOX inhibition from model to model, including stiffness measurements at the tissue level. Whilst for cholangiocarcinoma (EGI-1) and pancreatic adenocarcinoma models (KPC and mPDAC), LOX inhibition drastically decreases tumor stiffness, this effect is less marked in the breast adenocarcinoma model (MMTV-PyMT). Although collagen fiber curvature was affected in all models, changes in fiber orientation were only significant in KPC and mPDAC models. These changes can be explained by the fact that the tumor microenvironment and thus the basal collagen structure in each tumor model is different; hence, the inhibition of collagen crosslinking modifies the tumor collagen network to different extents. This emphasizes the need of developing reliable diagnostic markers, such as SWE stiffness mapping, based on a clear understanding of the tumor collagen network, in order to predict the response to ECM-targeting strategies.

In this study, LOX inhibition was performed in a preventive setting. We assume that it is during early tumor stages and the construction of fibrous stroma that LOX activity is the highest. In established tumors, with stroma having a significant level of crosslinked collagen, LOX inhibition would have a more marginal effect. Although not dealing with anti-PD-1 treatment, several articles are consistent with the notion that LOX inhibition is more effective in early tumor stages (*Nilsson et al., 2016*; *Miller et al., 2015*). For example, blocking LOX in combination with gemcitabine reduced metastases and increased survival of the mice when treatment was started in the early stages of the disease, but not at later stages (*Miller et al., 2015*).

It is well established that the number of T cells found within a tumor as well as their ability to migrate and reach cancer cells is key for an effective antitumoral response. These last few years a lot of efforts have been made in identifying cells and factors controlling the migration of T cells within tumors. The notion that prevailed is that growing tumors are composed of cells and factors, such as macrophages and hypoxia, hostile for T cells to migrate (*Peranzoni et al., 2018*; *Manaster et al., 2019*). The importance of the ECM in controlling the distribution and migration of T cells in tumors has also emerged. In human lung and ovarian tumors we found that T cells preferentially accumulate and migrate in stromal regions that exhibited a loose matrix architecture but not in dense regions (*Salmon et al., 2012*; *Bougherara et al., 2015*). Likewise, in aged skin, dermal fibroblasts harbor a phenotype similar to CAFs and produce ECM matrices that limit T cell displacements (*Kaur et al., 2019*). However, in triple-negative breast cancers and in pancreatic tumors, T cells were still found in dense networks of collagen fibers (*Li et al., 2019*; *Carstens et al., 2017*). Since most of the aforementioned studies were correlative, we decided in this study to specifically alter the ECM network by LOX inhibition and investigated the consequences on T cell motile behavior. Our results confirmed the importance of the ECM and tissue stiffness in controlling the migration of T cells in tumors. In particular, enhancements of T cell displacements were noticed in all tumor models under LOX inhibition. When we extended our analysis at the level of the individual mouse, we found that T cell intratumoral motility was inversely correlated to tumor stiffness as measured non-invasively using SWE (*Figure 6*), and this was true in all tested models. However, this relationship is not linear. In most models, T cells strongly decelerate when a threshold in stiffness is reached. In soft tumors such as those induced by LOX inhibition, T cells manage to migrate. Conversely, in stiff non-treated control tumors, T cell migration is impeded. These data fit well with results obtained in vitro in a range of 3D collagen matrices showing that T lymphocytes have the ability to adapt their morphology to

the structure of the tissue up to a certain limit (*Wolf et al., 2013*). In dense collagen matrices, T cell motility is halted. Our analysis supports the idea that elastography measurements could provide valuable companion markers to evaluate the need for anti-stromal strategy in order to normalize tumor stiffness and consequently improve T cell migration.

In this study, we did not take into consideration the possible effects of LOX inhibition on other determinants that could either positively or negatively regulate T cell migration. Along with tumor ECM, other cells and factors play important roles in controlling T cell migration. Hence, PD-1 blockade that does not affect collagen structure increases T cell migration within tumors (*Figure 7—figure supplement 1F*). It is well established that anti-PD-1 treatment is associated with the production of IFNγ and inflammatory chemokines (e.g. CXCL10) that are presumably responsible for enhancing T cell motility in tumors (*Peng et al., 2012*). A number of studies have shown that fibrotic tumors are, at the same time, enriched in cells and factors that are known to impede T cells from migrating (e.g. hypoxia and tumor-associated macrophages) (*Manaster et al., 2019*; *Maller et al., 2021*; *Deligne et al., 2020*). Therefore, a reduction in ECM content might normalize the tumor stroma rendering it more prone to stimulate T cell migration. The data in *Figure 7* showing a decrease in PMN and Treg and an increase in MHC class II + macrophages upon BAPN treatment support this assumption.

As for possible effects of LOX inhibition on tumor blood vessels, previous studies have reported reduced angiogenesis after LOX and LOX-like protein inhibition and an increased perfused vessel density in the case of overexpression of LOX (*Baker et al., 2013*; *Zaffryar-Eilot et al., 2013*). This could partly explain why we observe a significant increase in T cell infiltration in KPC tumors upon BAPN treatment. However, other recent studies argue the opposite as an increase in collagen crosslinking and matrix stiffness resulted in an increase in angiogenic sprouting. Conversely, the inhibition of collagen crosslinking in tumors resulted in reduced vascular density and permeability (*Bordeleau et al., 2017*).

Given the low efficacy of T cell-based immunotherapies in solid tumors, any method to increase its effect on tumor regression is of interest. With the exception of desmoplastic melanomas (*Eroglu et al., 2018*), features of wound healing and fibrosis are usually detrimental to anti-PD-1 responses (*Hugo et al., 2016*). Accordingly, a number of anti-fibrotic strategies have been recently implemented in combination with immune checkpoint inhibitors (*Jiang et al., 2017*). One of the most promising targets appears to be TGFβ. In preclinical mouse tumor models, TGFβ inhibition with immune checkpoint blockade induces complete and durable responses in otherwise unresponsive tumors (*Mariathasan et al., 2018*; *Tauriello et al., 2018*). However, due to TGFβ pleiotropic effects, concerns regarding the blockade of this cytokine arose.

Our study indicates that LOX represents another valuable target as its inhibition in the transplanted KPC model increases the efficacy of anti-PD-1 treatment, while monotherapy with either agent alone is ineffective. Moreover, the combination treatment was associated with a tumor microenvironment shifted toward antitumoral effector cells and components, whereas immunosuppressive cells were reduced.

Although the clinical use of BAPN has been impeded by concerns regarding toxicities, other strategies to inhibit LOX in cancer and fibrotic disease patients are currently ongoing (*Lampi and Reinhart-King, 2018*). Our work confirms LOX as a molecular target to improve T cell migration dynamics as well as to ameliorate the immunosuppressive microenvironment. It paves the way for clinical trials combining LOX inhibitors with PD-1/PD-L1 blockade, possibly in biomarker-selected cohorts of patients with high tumor stiffness evaluated with non-invasive imaging approaches such as SWE.

## Materials and methods

**Key resources table**

| Reagent type (species) or resource | Designation | Source or reference | Identifiers | Additional information |
|---|---|---|---|---|

*Continued on next page*

*Continued*

| Reagent type (species) or resource | Designation | Source or reference | Identifiers | Additional information |
|---|---|---|---|---|
| Strain, strain background (*M. musculus*) | FVB/N | Janvier Labs | FVB/N | |
| Strain, strain background (*M. musculus*) | C57Bl/6J | Janvier Labs | C57Bl/6J | |
| Strain, strain background (*M. musculus*) | MMTV-PyMT | *Guy et al., 1992* | | Spontaneous mammary carcinoma mouse model |
| Strain, strain background (*M. musculus*) | NMRI-nu (nu/nu) | Envigo | NMRI nude | HsdCpb:NMRI-Foxn1nu |
| Cell line (*Homo-sapiens*) | EGI-1 | German Collection of Microorganisms and Cell Cultures | ACC385 | Human bile duct carcinoma cell line |
| Cell line (*M. musculus*) | KPC | Kind gift from Corinne Bousquet, Université Toulouse III | | Cell line derived from a pancreatic tumor obtained from $Kras^{LSL\_G12D}$, $Trp53^{R172H/+}$ mice. |
| Cell line (*M. musculus*) | mPDAC | Kind gift from Douglas Hanahan, Swiss Institute for Experimental Cancer Research | | Cell line derived from a pancreatic tumor obtained from $Kras^{LSL\_G12D}$, $Trp53^{R172H/+}$, $Cdkn2a^{-/+}$ mice. |
| Cell line (*M. musculus*) | MET-1 | Kind gift from Robert Cardiff, the University of California Research (*Borowsky et al., 2005*) | | Cell line derived from a mammary carcinoma in FVB/N-Tg (MMTV-PymT). |
| Antibody | Anti-CD11b (monoclonal rat anti-mouse/human, clone M1/70) | Biolegend | 101201 RRID:AB_312784 | (10 μg/mL) |
| Antibody | Anti-CD11c (monoclonal rat anti-mouse, clone N418) | Biolegend | 117333 RRID:AB_11204262 | (10 μg/mL) |
| Antibody | Anti-CD45 (monoclonal rat anti-mouse, clone 30-F11) | Biolegend | 103127 RRID:AB_493714 | (10 μg/mL) |
| Antibody | Anti-Ly6C (monoclonal rat anti-mouse, clone HK1.4) | Biolegend | 128025 RRID:AB_10643867 | (10 μg/mL) |
| Antibody | Anti-Ly6G (monoclonal rat anti-mouse, 1A8) | Biolegend | 127633 RRID:AB_2562937 | (10 μg/mL) |
| Antibody | Anti-CD4 (monoclonal rat anti-mouse, clone GK1.5) | BD Biosciences | 563050 RRID:AB_2737973 | (10 μg/mL) |
| Antibody | Anti-CD8a (monoclonal rat anti-mouse, clone 53–6.7) | eBioscience | 46-0081-82 RRID:AB_1834433 | (10 μg/mL) |
| Antibody | Anti- TCRβ (monoclonal hamster anti-mouse, clone H57-597) | BD Biosciences | 562840 RRID:AB_2687544 | (10 μg/mL) |
| Antibody | Anti-Nkp46 (monoclonal rat anti-mouse, clone 29A1.4) | eBioscience | 50-3351-82 RRID:AB_10598664 | (10 μg/mL) |

*Continued on next page*

*Continued*

| Reagent type (species) or resource | Designation | Source or reference | Identifiers | Additional information |
|---|---|---|---|---|
| Antibody | Anti-PD-1 (monoclonal rat anti-mouse, clone 29F.1A12) | Biolegend | 135221 RRID:AB_2562568 | (10 µg/mL) |
| Antibody | Anti-MHC II (monoclonal rat anti-mouse, clone M5/114.15.2) | Biolegend | 107645 RRID:AB_2565977 | (10 µg/mL) |
| Antibody | Anti-CD80 (monoclonal hamster anti-mouse, clone 16-10A1) | Biolegend | 104705 RRID:AB_313126 | (10 µg/mL) |
| Antibody | Anti-CD64 (monoclonal mouse anti-mouse, clone X54-5/7.1) | Biolegend | 139303 RRID:AB_10613467 | (10 µg/mL) |
| Antibody | Anti-CD206 (monoclonal rat anti-mouse, clone CO68C2) | Biolegend | 141721 RRID:AB_2562340 | (10 µg/mL) |
| Antibody | Anti-F4/80 (monoclonal rat anti-mouse, clone CI:A3-1) | BIO-RAD | MCA497SBV670 RRID:AB_323806 | (10 µg/mL) |
| Antibody | Anti-FoxP3 (monoclonal rat anti-mouse, clone MF-14) | Biolegend | 126405 RRID:AB_1089114 | (10 µg/mL) |
| Antibody | Anti-GrzmB (monoclonal rat anti-mouse, clone QA16A02) | Biolegend | 372207 RRID:AB_2687031 | (10 µg/mL) |
| Antibody | Anti-Fibronectin (monoclonal mouse anti-human/mouse, clone HFN7.1) | Acris Antibodies | AM00389AF-N RRID:AB_981328 | (10 µg/mL) |
| Antibody | Anti-Podoplanin (monoclonal Hamster anti- mouse, clone 8.1.1) | Biolegend | 127407 RRID:AB_2161929 | (10 µg/mL) |
| Antibody | Anti-EpCAM (monoclonal mouse anti-human, clone 9C4) | Biolegend | 324219 RRID:AB_11124342 | (10 µg/mL) |
| Antibody | Anti-EpCAM (monoclonal rat anti-mouse, clone G8.8) | Biolegend | 118225 RRID:AB_2563983 | (10 µg/mL) |
| Antibody | Anti-PD-1 (monoclonal rat anti-mouse, clone RMP1-14) | BioXcell | BE0146 RRID:AB_10949053 | (200 µg) |
| Peptide, recombinant protein | Human IL-7 | Miltenyi Biotec | 130-095-361 | |
| Peptide, recombinant protein | Human IL-15 | Miltenyi Biotec | 130-095-762 | |
| Commercial assay or kit | Milliplex map Mouse cytokine/chemokine detection | Merck Millipore | MCYTOMAG-70K | |
| Commercial assay or kit | Human CD8 + T Cell Isolation Kit | Stemcell technologies | 17953 | |
| Commercial assay or kit | CD8a + T Cell Isolation Kit, mouse | Miltenyi Biotec | 130-104-075 | |
| Commercial assay or kit | Human T cell activation kit, TransAct | Miltenyi Biotec | 130-111-160 | |

*Continued on next page*

*Continued*

| Reagent type (species) or resource | Designation | Source or reference | Identifiers | Additional information |
|---|---|---|---|---|
| Commercial assay or kit | Dynabeads Mouse T-Activator CD3/CD28 | ThermoFischer | 11456D | |
| Chemical compound, drug | beta-aminopropionitrile (BAPN) | Sigma | A3134 | (3 mg/mL) |
| Chemical compound, drug | CellTrace Calcein Red-Orange | ThermoFischer | C34851 | (125 nM) |
| Software, algorithm | Image J/FIJI | National Institutes of Health | | imagej.net |
| Software, algorithm | Imaris 7.4 | Oxford Instruments | | imaris.oxinst.com |
| Software, algorithm | Prism GraphPad Software version 8 | GraphPad | | graphpad.com |
| Software, algorithm | FlowJo | FlowJo, LLC | | flowjo.com |

## Cell culture

The KPC cell line was derived from a pancreatic tumor obtained from $Kras^{LSL\_G12D}$, $Trp53^{R172H/+}$ mice (C57BL/6 background, a generous gift from Corinne Bousquet, Université Toulouse III).

The pancreatic cancer cell line mPDAC was derived from a tumor obtained from $Kras^{LSL\_G12D}$, $Trp53^{R172H/+}$, $Cdkn2a^{-/+}$ mice (obtained from Douglas Hanahan, Swiss Institute for Experimental Cancer Research).

The MET-1 cell line was derived from a mammary carcinoma in FVB/N-Tg (MMTV-PymT) (obtained from Robert Cardiff, the University of California, Davis; *Borowsky et al., 2005*).

EGI-1 cells, derived from the extrahepatic biliary tract, were obtained from the German Collection of Microorganisms and Cell Cultures (DSMZ, Germany).

mPDAC and KPC cells were cultured in DMEM supplemented with 10% fetal bovine serum (FBS), penicillin/streptomycin, L-glucose, and sodium pyruvate (Gibco). EGI-1 cells were cultured in DMEM supplemented with 1 g/L glucose, 10 mmol/L HEPES, 10% FBS, antibiotics (100 UI/mL penicillin and 100 µg/mL streptomycin), and antimycotic (0.25 µg/mL amphotericin B) (Invitrogen).

Each cell line was thawed from lab-frozen stock which were generated from early passages and utilized for each experiment within 4 weeks culture. All the cell lines utilized were mycoplasma free determined by qPCR analysis.

Human CD8+ T cells were isolated from cytapheresis rings (obtained from Establissement Français du Sang) using EasySep Human CD8+ T Cell Isolation Kit (Stem Cell Technologies), following manufacturer's protocol. They were cultured in RPMI supplemented with 3% Human AB serum, T cell TransAct (Miltenyi Biotech), 155 U/mL of human IL-7 (Miltenyi Biotec), and 290 U/mL of human IL-15 (Miltenyi Biotech). Two to three days after activation, the cell culture media was changed using the same recipe without TransAct. At day 7 after activation, T cells were used for migration experiments.

Mouse CD8+ T cells were isolated from FVB mouse spleen and lymph nodes using CD8α+ T Cell Isolation Kit (Miltenyi Biotech) following the manufacturer's protocol. Isolated T cells were activated using the Dynabeads Mouse T-Activator CD3/CD28 for T-Cell Expansion and Activation (Thermo Fischer) following the manufacturer's protocol. On 7 seven after activation, T cells were used for migration experiments.

## In vivo studies

Mouse tumor models used in this study were as follows:

### EGI-1 subcutaneous model

$2 \times 10^6$ cells suspended in 60 μL of PBS were mixed with 60 μL of Matrigel growth factor reduced (Corning) and implanted subcutaneously into the flank of 5-week-old female NMRI-nu (nu/nu) mice (Envigo, France).

### KPC subcutaneous model

$3 \times 10^6$ cells suspended in 50 μL of PBS were mixed with 50 μL of Matrigel growth factor reduced (Corning) and implanted subcutaneously into the flank of 6-week-old female C57BL/6J mice (Janvier, France).

### MET-1 subcutaneous model

$10^6$ cells suspended in 50 μL of PBS were injected subcutaneously into the flank of a 6-week-old female FVB mouse (Janvier France).

### MMTV-PyMT model

FVB MMTV-PyMT model is maintained at the Cochin Institute specific-pathogen-free- animal facility in accordance with the University Paris Descartes ethical guidelines.

### mPDAC orthotopic model

$10^3$ cells suspended in 50 μL of PBS were injected orthotopically in the pancreas of 6-week-old FVB/n mice (Janvier, France). Tumor growth was followed through ultrasound imaging using a VEVO2100 (Visualsonics).

Tumor growth was followed with a caliper, and tumor volume (V) was calculated as follows: xenograft volume = $(xy^2)/2$ where x is the longest and y, the shortest of two perpendicular diameters.

For LOX inhibition, animals were treated with BAPN (3 mg/mL, Sigma) in the drinking water, which was changed twice per week. In all implantable models (KPC, mPDAC, MET-1, and EGI-1), animals were treated from the day of the tumor implantation until the end of the study. For the MMTV-PyMT model, animals were treated at 10 weeks of age, approximately the moment the tumor started developing.

Experiments combining LOX inhibition and anti-PD1 were conducted in the KPC model. Four i.p. injections of 200 μg of anti–PD-1 antibody (RMP1-14 clone; BioXcell) or isotype control (rat IgG2a; BioXCell) were started when tumors reached 80–100 mm³ size. Injections were performed every 4 days.

All animal experiments were performed in agreement with institutional animal use and care regulations after approval by the animal experimentation ethics committee of Paris Descartes University (CEEA 34, 16–063).

## Shear wave elastography

SWE measurements were performed every 3–4 days during the entire follow-up of the tumor growth. Images were acquired with the ultrasound device Aixplorer (SuperSonic Imagine, Aix-en-Provence, France) using a 15 MHz superficial probe dedicated to research (256 elements, 0.125 μm pitch).

The mice were anesthetized with 2% isoflurane, and their body temperature was maintained at a physiological level using a heating plate. B-mode images and SWE images were acquired simultaneously. The B-mode image allowed us to manually determine the region of interest (ROI) corresponding to the tumor contours. SWE mode was performed using the penetration mode with a color scale ranging from 0 (blue) to 40 kPa (red); this cut-off was chosen based on the results from a previous study performed by our group (*Marangon et al., 2017*). The area, the diameter, and a set of stiffness values (mean, minimum, maximum, and standard deviation) were recorded for the ROI as previously defined. SWE images were also analyzed using an in-house MATLAB code to recover the stiffness map. Once the stiffness map was recovered, the percentage of pixels within the ROI with stiffness values > 40 kPa with respect to the total number of pixels in the region was calculated in order to analyze the percentage of stiff regions. The normalized tumor stiffness was calculated by normalizing each time point measurement with that of the initial time point (Stiffness $t_n$/Stiffness $t_0$).

## Tumor slice imaging

Tumor slices were prepared following the protocol described previously (*Peranzoni et al., 2018*). Briefly, samples were embedded in 5% low-gelling-temperature agarose (type VII-A; Sigma-Aldrich) prepared in PBS. Slices (350 µm) were cut with a vibratome (VT 1000S; Leica) in a bath of ice-cold PBS.

To evaluate T cell migration in the EGI-1 model, activated human CD8+ T cells were first labeled with CellTrace Calcein red-orange dye (ThermoFischer). Briefly, T cells were incubated with 125 nM Calcein in HBSS solution for 10 min at 37°C. The staining reaction was stopped by adding cold HBSS supplemented 2% SAB. Cells were then pelleted and diluted to an appropriate concentration in phenol red-free RPMI media. $2 \times 10^5$ cells suspended in 50 µL of phenol red-free RPMI media and added on 350 µm tumor slices placed on top of 0.4 µm organotypic culture inserts (Millicell; Millipore) in 35 mm Petri dishes containing 1.1 mL RPMI 1640 without Phenol Red. The tumor slices were then incubated for 30 min at 37°C and 5% $CO_2$. The slices were then washed to remove all cells that had not infiltrated the slice and stained for 15 min at 37°C with the following antibodies: BV421–anti-human EpCAM (9C4 clone; BioLegend) and anti-human/mouse fibronectin at 10 µg/mL (HFN7.1 clone; Acris antibodies). The same protocol was followed to evaluate T cell migration in MMTV-PyMT cells except that activated murine CD8+ T cells were used.

To evaluate resident T cell migration in KPC and mPDAC model, tumor slices were then transferred to 0.4 µm organotypic culture inserts (Millicell; Millipore) in 35 mm Petri dishes containing 1.1 mL phenol red-free RPMI media. Live vibratome sections were stained with BV421 anti-mouse EpCAM (G8.8 clone; BD Biosciences), PerCP-e710 anti-mouse CD8a (53–6.7 clone, eBioscience), PE anti-podoplanin (8.1.1 clone; BioLegend), and anti-human/mouse fibronectin (HFN7.1 clone; Novus Biologicals).

T cells were imaged with a DM500B upright microscope equipped with an upright spinning disk confocal microscope (Leica) equipped with a 37°C thermostatic chamber. For dynamic imaging, tumor slices were secured with a stainless steel slice anchor (Warner Instruments) and perfused at a rate of 0.8 mL/min with a solution of RPMI without Phenol Red, bubbled with 95% $O_2$ and 5% $CO_2$. Ten minutes later, images from a first microscopic field were acquired with a 25× water immersion objective (20×/0.95 N.A.; Olympus). For four-dimensional analysis of cell migration, stacks of 10–12 sections (z step = 5 µm) were acquired every 30 s for 20 min at depths up to 80 µm. Regions were selected for imaging when tumor parenchyma, stroma, and T cells were simultaneously present in the same microscopic field. For most of the tumors included in the study, between two and four microscopic fields were selected for time-lapse experiments.

## Dynamic imaging analysis

A 3D image analysis was performed on x, y, and z planes using Imaris 7.4 (Oxford Instruments). First, superficial planes from the top of the slice to 15 µm in depth were removed to exclude T cells located near the cut surface. Cellular motility parameters were then calculated. Tracks of >10% of the total recording time were included in the analysis.

## SHG microscopy

The images were obtained using an inverted stand Leica SP5 microscope (Leica Microsystems GmbH, Wetzlar, Germany) coupled with a femtosecond Ti:sapphire laser (Chameleon, Coherent, Saclay, France) tuned at a wavelength of 810 or 850 nm for all experiments. The laser beam was circularly polarized and a Leica Microsystems HCX IRAPO 25×/0.95 W objective was used. SHG (collagen structure) signal was detected in epi-collection through a 405/15 nm bandpass filters, respectively, by NDD PMT detectors (Leica Microsystems) with a constant voltage supply, at constant laser excitation power, allowing direct comparison of SHG intensity values. All images were then analyzed using CT-FIRE software to obtain the width and the length of the collagen fibers. To calculate the curvature ratio, line regions were drawn along the length of the fibers (A) and along the linear distance between the start and the end of the fibers (B), the curvature ratio was calculated CR = A/B. For each image at least, 15 fibers were analyzed. Fiber alignment was determined using the directionality plugin in Image J. The fiber alignment was defined by the CV of the angle for all fibers per image. The smaller the CV is, the more aligned the fibers are.

### Detection of collagen crosslinks using two-photon microscopy

An upright stand Leica SP5 multiphoton microscope (Leica Microsystems) was used to assess PYD and DPD natural fluorescence within tumor slices of KPC tumors. Samples were illuminated at a wavelength of 720 nm using a Ti:Sa Chameleon Ultra II (Coherent, Saclay, France) as a laser source. Fluorescence emissions were measured at a wavelength of $400 \pm 25$ nm. Mean fluorescent signals were measured in SHG-positive regions. The background noise of each region was subtracted.

### Histology

For most of the tumors, half of the biopsy was fixed overnight at 4°C in a periodate–lysine–paraformaldehyde solution (0.05 M phosphate buffer containing 0.1 M L-lysine [pH 7.4], 2 mg/mL $NaIO_4$, and 10 mg/mL paraformaldehyde). After fixation, tumors were dehydrated in graded solutions of ethanol and embedded in paraffin. Five micrometer tissue sections were stained with HES and Sirius Red. For Sirius Red-stained slices, linear polarized light or bright field microscopy was performed using full-field microscopy (Statif Axio Observer Z.1, Zeiss) equipped with a linear polarizer and a 20× dry objective (Plan Achromatic [NA = 0.7]). Under polarized light, thin fibers show a greenish-yellow birefringence, whilst thicker and densely packed fibers give an orange-red birefringence. The percentage of Sirius Red staining defined the amount of thick and densely packed fibers, to do so images were split in the RGB channel and the signal in the red channel was quantified.

### Atomic force microscopy and shear rheometry

Mechanical measurements of the control and BAPN mice tumors were performed ex vivo at the nanoscopic and macroscopic scale using AFM and shear rheometer.

AFM experiments were made maximally 3 hr after sample thawing, and tissues were stored and kept in culture medium during the experiment at room temperature. Millimeter-scale samples of mice KPC model tumors were measured with a JPK Bruker NanoWizard 4 BioScience atomic force microscope working in the force spectroscopy mode. Force vs. indentation curves were collected using a silicon nitride cantilever with a spring constant of $0.6 \ N/m$ and a 4.5 µm diameter bead attached. Each sample was indented in multiple locations to account for possible heterogeneity in tissue mechanical properties. Briefly, up to nine maps consisting of 64 points corresponding to the scan area of $10 \times 10$ µm (for one map) per sample were made. Final Young's modulus values were derived from the Hertz–Sneddon model applied to force vs. indentation curves (*Pogoda et al., 2012*), assuming the spherical shape of the probe and Poisson's ratio equal 0.5. Distributions of Young's modulus values for each control and treated sample, as well as the mean values along with standard deviations, were prepared.

For the macroscopic rheological tests, HAAKE Rheostress 6000 rheometer (Thermo Fisher Scientific), fitted with an 8 mm diameter parallel plate system, was used. Tissues were firstly cut into disk-shaped samples using an 8 mm punch. To avoid tissue slippage during the tests, samples were arranged in sand paper glued inside the Petri dishes and fixed to the rheometer bottom plate. Rheological experiments were made maximally 3 hr after tissue thawing, and all the samples were kept in humid conditions during the experiment. The rheological evaluation consisted of the oscillating shear deformation with 2% shear amplitude and frequency of 1 Hz. Final results are presented as the mean values of storage modulus ($G$) of control and treated tumors ± standard deviation (SD) values.

### Flow cytometry

Immune cells in tumors were stained as described previously (*Peranzoni et al., 2018*). In brief, tumors were mechanically dissociated and digested for 45 min at 37°C in RPMI 1640 with 37.5 µg/mL Liberase TM (Roche) and 8,000 U/mL DNase I from bovine pancreas (Merck Millipore). The resulting digestion was filtered through a 70 µm cell stainer and centrifuged. Red blood cell lysis with ACK buffer was performed on the remaining pellet and subsequently filtered on a 40 µm cell strainer. The cell suspension was then rinsed in PBS and stained in 96-well round-bottom plates with a LIVE/DEAD Fixable Blue Dead Cell Stain Kit (Invitrogen) for 20 min at 4°C. Cells were then washed and stained with Abs against surface proteins at a concentration of 10 µg/mL for 20 min at 4°C.

The anti-mouse antibodies used were the following:

| Antigen | Fluorophore | Clone | Company |
|---------|-------------|-------|---------|
| CD11b | PE-Cy7 | M1/70 | Biolegend |
| CD11b | BV421 | M1/70 | BD Biosystems |
| CD11c | BV605 | N418 | Biolegend |
| CD45 | AF700 | 30-F11 | Biolegend |
| Ly-6C | APC-Cy7 | HK1.4 | Biolegend |
| Ly-6G | BV510 | 1A8 | Biolegend |
| CD4 | BV 711 | GK1.5 | BD Biosciences |
| CD8a | PerCP-e710 | 53–6.7 | eBioscience |
| TCRβ | BV605 | H57-597 | BD Biosciences |
| Nkp46 | eFluor660 | 29A1.4 | eBioscience |
| PD-1 | BV421 | 29F.1A12 | Biolegend |
| MHC II | BV785 | M5/114.15.2 | Biolegend |
| CD80 | FITC | 16-10A1 | Biolegend |
| CD64 | PE | X54-5/7.1 | Biolegend |
| CD206 | BV605 | CO68C2 | Biolegend |
| F4/80 | StarBright Violet 670 | CI:A3-1 | BIO-RAD |

After surface staining, cells were fixed with BD fixation and permeabilization solution for 20 min at 4°C. In the case of intracellular staining, cells were incubated overnight with the following anti-mouse antibodies. All antibodies were used at a concentration of 10 µg/mL:

| Antigen | Fluorophore | Clone | Company |
|---------|-------------|-------|---------|
| FoxP3 | AF488 | MF-14 | Biolegend |
| GrzmB | PE | QA16A02 | Biolegend |

After washing in PBS, cells were resuspended in PBS 2% FBS and analyzed with a BDFortessa flow cytometer (BD Bioscience). Data were analyzed by FlowJo software.

### Cytokine detection in tumor slice supernatants

Fresh KPC tumor slices were prepared as previously described and kept at 37°C in 24-well plates with 0.5 mL RPMI per well. Four to five slices were put in culture for each mouse. Eighteen hours later supernatants were collected and centrifuged at $300 \times g$ to eliminate suspension cells. Cell-free supernatants were frozen and stored at 80°C. Granzyme B, TNFα, and RANTES (CCL5) release was assayed by Luminex technology (Bio-Plex 200 from Bio-Rad) with a customized Milliplex kit (Merck Millipore).

### Statistical analysis

Results were analyzed using the GraphPad Prism 5.0 statistical software. Data are shown as means ± standard error of the mean (SEM). For comparisons between two groups, parametric Student's t-test or non-parametric Mann–Whitney test were used. For comparisons between more than two groups, a parametric one-way analysis of variance (ANOVA) test was followed by a posteriori Kruskal–Walis test.

### Acknowledgements

This study was supported by grants from the French Ligue Nationale Contre le Cancer (Equipes labellisées) (ED), Plan Cancer (Tumor heterogeneity and ecosystem program) (ED), CARPEM (Cancer Research for Personalized Medicine) (ED), European Union's Horizon 2020 research and innovation program under grant agreement N°685795 (NoCanTher) (FG). ANB received a PhD fellowship by the Institute Thematique Multi-organisms (ITMO) Cancer, the doctoral school Frontières du Vivant

(FdV) – Programme Bettencourt and the Fondation ARC pour la recherche sur le cancer. LF and JV are members of the European Network for the Study of Cholangiocarcinoma (ENSCCA) and participate in the initiative COST action EURO-CHOLANGIO-NET granted by the COST Association (CA18122). JV was funded by the LABEX Plas@par project and received financial state aid managed by the Agence Nationale de la Recherche, as part of the programme 'Investissements d'avenir' (ANR-11-IDEX-0004–02). We would like to thank the staff of the IMAG'IC, CYBIO, PIV, and HistIM facilities of the Cochin Institute for their advice during this study. IMAG'IC facility is supported by the National Infrastructure France BioImaging (grant ANR-10-INBS-04) and IBISA consortium. We acknowledge Tatiana Ledent from Housing and experimental animal facility (HEAF), and Fatiha Merabtene and Brigitte Sohlonne from the histomorphology platform, Centre de Recherche Saint-Antoine (CRSA). Dr. Joanna Mystkowska and Dawid Lysik (a PhD student) from the Department of Materials Engineering and Production, Faculty of Mechanical Engineering from Bialystok University of Technology for facilitating the measurements using the rheometer.

## Additional information

### Funding

| Funder | Grant reference number | Author |
|---|---|---|
| Ligue Contre le Cancer | EL2020.LNCC/EmD | Alba Nicolas-Boluda<br>Lene Vimeux<br>Sarah Barrin<br>Chahrazade Kantari-Mimoun<br>Emmanuel Donnadieu |
| Institut National Du Cancer | Program HTE | Alba Nicolas-Boluda<br>Lene Vimeux<br>Sarah Barrin<br>Chahrazade Kantari-Mimoun<br>Emmanuel Donnadieu |
| European Commission | 685795 | Florence Gazeau |
| Agence Nationale de la Recherche | 11-IDEX-0004-02 | Javier Vaquero |
| Fondation pour la Recherche Médicale | 202003010517 | Laura Fouassier |

The funders had no role in study design, data collection and interpretation, or the decision to submit the work for publication.

### Author contributions

Alba Nicolas-Boluda, conceived the project, designed and performed the experiment, analyzed the data, and wrote the manuscript; Javier Vaquero, contributed to the in vivo experimental section; Lene Vimeux, performed imaging experiments and analyzed the data; Thomas Guilbert, assisted with two-photon experiments and data analysis during the revision process; Sarah Barrin, Chahrazade Kantari-Mimoun, assisted with imaging experiments and data analysis; Matteo Ponzo, Laura Fouassier, assisted with in vivo mouse experiments and data analysis; Gilles Renault, assisted with shear wave elastography experiments and data analysis; Piotr Deptula, Katarzyna Pogoda, Robert Bucki, performed Atomic Force Microscopy experiments and data analysis; Ilaria Cascone, José Courty, assisted with in vivo mouse experiments; Florence Gazeau, conceived and supervised the project, designed the experiment, and wrote the manuscript; Emmanuel Donnadieu, conceived and supervised the project, designed the experiment, analyzed the data, and wrote the manuscript

### Author ORCIDs

Laura Fouassier (iD) https://orcid.org/0000-0001-6377-5610
Emmanuel Donnadieu (iD) https://orcid.org/0000-0002-4985-7254

### Ethics

Animal experimentation: All animal experiments were performed in agreement with institutional animal use and care regulations after approval by the animal experimentation ethics committee of Paris Descartes University (CEEA 34, 16-063).

### Decision letter and Author response

Decision letter https://doi.org/10.7554/eLife.58688.sa1
Author response https://doi.org/10.7554/eLife.58688.sa2

## Additional files

### Supplementary files

• Supplementary file 1. Summary of relevant preclinical models for solid tumors used in this study.

• Supplementary file 2. Tumor stiffness and tumor architecture parameters for EGI-1, KPC, MMTV-PyMT, mPDAC and MET-1 models. Mean stiffness of tumors with volume > 600 mm$^3$ measured with SWE. Percentage of stiff regions of tumors with volume > 600 mm$^3$. Tumor architecture was characterized by the percentage of the tumor covered by the stromal compartment, estimated from HES images; collagen fiber width and length, calculated from SHG images; percentage of red-orange birefringent fibers combining Red Sirius staining and polarized microscopy, orange-red fibers correspond to thick and packed regions. EGI-1 (n = 30 mice/group from three independent experiments); KPC (n = 34 mice/group from three independent experiments); MMTV-PyMT (n = 5 mice/group, 10 tumors per mouse from two independent experiments); mPDAC (n = 7 mice/group from two independent experiments); and MET-1 (n = 12 mice/group from two independent experiments).

• Supplementary file 3. Collagen fiber width and length in EGI-1, MMTV-PyMT, mPDAC and KPC tumor models in control and BAPN-treated conditions. Results are shown as mean ± SD.

• Supplementary file 4. T cell migration in the different tumor models and effect of LOX inhibition. Migration of activated PBT overlaid onto fresh tumor slices was analyzed in EGI-1 and MMTV-PyMT model, whilst resident tumor infiltrated T lymphocyte migration was analyzed in mPDAC and KPC models (* p-value<0.05, ***p-value<0.001, Students' t-test, n = 3–12 mice/group from three experiments, 70–250 T cells per slice analyzed). T cell migration was analyzed at the endpoint of the experiment: day 30 for EGI-1, day 40 for KPC, day 55 for MMTV-PyMT and day 21 for mPDAC. T cell infiltration (CD8/mm$^2$) calculated from immunofluorescence images. Results are shown as mean ± SD. Illustrative images of T cell migration tracks in EGI-1 tumor model. Tumor stroma (fibronectin) in red, tumor cells (EpCAM) in blue and T cells (Calcein) in green. Tracks are color-coded to illustrate track displacement. Scale bar = 100 µm. See also *Videos 1* and *2*. TILs: tumor-infiltrating T lymphocytes.

• Transparent reporting form

### Data availability

Relevant source data for all figures and supplement figures have been uploaded as Excel files.

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
