## [Decision Letter]

**Acceptance summary:**

An elevated percentage of patients with solid tumors fail to respond to immunotherapies. It is due not only to the 'exhausted' phenotype of tumor-infiltrating T cells but also to the fact that several fibrotic tumors show physical resistance to T cell infiltration. The present study highlights the detrimental impact of collagen fibrils architecture on the migratory behavior of anti-tumoral T cells in fresh human tumor explants, and specifically explores the consequences of 'softening' the extracellular matrix by inhibition of the lysyl oxidase (LOX), a copper-dependent enzyme that is responsible for the crosslinking of collagen molecules into fibers and is overexpressed in many metastatic tumors. The study shows that LOX inhibition has different mechanical modulating effects depending on the ECM architecture and results in significant improvement in T cell mobility. Importantly, combining LOX inhibition and PD-1 blockade treatment increases effector CD8 T cell accumulation in tumors and significantly delays tumor progression in a pancreatic cancer model.

**Decision letter after peer review:**

Thank you for submitting your article "Tumor stiffening reversion through collagen crosslinking inhibition improves T cell migration and anti-PD-1 treatment" for consideration by *eLife*. Your article has been reviewed by 3 peer reviewers, and the evaluation has been overseen by a Reviewing Editor and Päivi Ojala as the Senior Editor. The reviewers have opted to remain anonymous.

The work analyzes the stiffness and collagen distribution in different tumor models implanted in mice and shows that treatment with an inhibitor of collagen crosslinking correlates with changes in their stiffness. This results in a change in the motility of resident T cells. The inhibitor of collagen crosslinking increases the number of tumor-infiltrating CD8^+^ T cells and leads to increased efficacy of anti-PD-1 blockade on tumor growth. The reviewers have discussed the reviews with one another and the Reviewing Editor and their views concur. Although the work has potential for publication in *eLife*, it requires essential additional data and statistics to support the central claims of the paper. Each reviewer raised substantive concerns (see below) that need to be resolved experimentally. To quote a few, you should provide a measurement of the collagen crosslinking in mice treated by BAPN to confirm that this drug has the expected effects. The combined BAPN plus anti-PD-1 therapy needs also to be confirmed in another model. Measurements of stiffness, collagen structure and T cell speed should be provided for all treatment conditions (control, LOXi, PD1i and combo) rather than just for LOX inhibition. Importantly, several important conclusions are based on inadequate sample size to be conclusive (see below). Along that line, the number of mice and tumor cells plus corresponding statistics need to be indicated in all the figures.

*Reviewer #1:*

In their article entitled "Tumor stiffening reversion through collagen crosslinking inhibition improves T cell migration and anti-PD-1 treatment" Alba Nicolas-Boluda and co-authors analyze the stiffness and collagen distribution in different tumor models implanted in mice. They show that treatment with an inhibitor of collagen crosslinking modifies the collagen network in these tumors and that this correlates with changes in their stiffness. They then analyze the motility of T cells in the different models and show that this motility is modified by the treatment and correlates with the stiffness of the tumor. In the last part of their study, the authors show that treatment of the mice with the inhibitor of collagen crosslinking changes the immune infiltrates in the tumors characterized by a more abundant presence of CD8^+^ T cells. They finally show that interfering with collagen stabilization leads to increased efficacy of anti-PD-1 blockade on tumor growth.

Relevance of the study:

T cells are excluded from a large proportion of solid tumor. This represents an obstacle to T-cell-based immunotherapies. The authors make the hypothesis that this can be, at least partly, due to the organization of the ECM in the tumor that would oppose physical resistance to the infiltration and migration of T cells. The results are sound and important for the community since 1) they describe thoroughly some of the mechanical aspects of several models used in the literature, 2) they thoroughly analyzed the effect of an inhibitor of collagen crosslinking on these mechanical properties 3) study the effects of these modifications in T cell motility and 4) test in one tumor model the effects of the combination of an inhibitor of collagen crosslinking with anti-PD1 immunotherapy. The results are convincing and I only have minor concerns.

In the first part of their study, the authors analyze the structure heterogeneity of 5 different carcinomas, i.e. subcutaneous model of cholangiocarcinoma (EGI-1), subcutaneous (MET-1) and transgenic model (MMTV-PyMT) of mouse breast carcinoma, orthotopic (mPDAC) and subcutaneous (KPC) models of mouse pancreatic ductal adenocarcinoma.

They measure the tumor stiffness during tumor growth using Shear Wave Elastography (SWE) and analyze the organization of the collagen fibers in these models. To my knowledge, this represents the first characterization of different tumor models classically used to study tumor immunity and is thus very useful for the scientific community. In particular, the authors show a correlation between high tumor stiffness and accumulation of thick and densely packed collagen fibers.

Minor modifications: The authors should indicate more clearly the number of mice and tumors investigated.

In the second part of their study, the authors treat the mice with beta-aminopropionitrile (BAPN), an inhibitor for LOX enzymatic activity in the drinking water and analyze the stiffness of tumors and collagen fiber organization in tumors. They show the heterogeneity of response in the different models in both stiffness modulation and collagen fibers remodeling. Mostly this treatment reduces the stiffness of tumors without affecting their growth.

Minor modifications: The authors should clarify how "normalized tumor stiffness" indicated in the legend of figure 2 is calculated. Indeed, this is an important point since tumor stiffness is associated to the sizes of tumors. Moreover, they should also indicate more clearly the number of mice and tumors investigated. Concerning collagen fibers orientation, authors should use a dot plot representation instead of bar histograms in order to show the distribution in the different tumors.

The authors then analyze how BAPN treatment modifies the migration of T lymphocytes in the tumors. Because of the different models used, the authors either added activated purified T cells from human donors (EGI-1model), or mouse activated T cells (MMTV-PyMT tumor model) or followed the motility of human resident T cells in mPDAC and KPC mice tumor models. Although the models are very different, the correlation between tumor stiffness and T cell speed and T cell displacement is specially striking in tumors from BAPN treated mice. It seems that T cell motility responds to two different regimens in tumor from untreated or BAPN treated mice. This might be due to difference of stiffness in untreated and treated mice but might also results from another parameter.

Minor modifications: The authors should discuss this point. Indeed, the main conclusion of their work and short title of their study is that the main parameter involved in T cell motility and access to the tumor is tumor stiffness but then the slopes should be the same as in the spontaneous MMTV-PyMT tumor model. There are probably other parameters involved in the regulation.

The authors then investigate the effect of BAPN treatment of tumor bearing mice on response to PD-1 immunotherapy. They perform experiments in KPC tumor bearing mice and show that BAPN treatment alone significantly decreases the number of neutrophils, increases the presence of MHCII+ TAMs. Yet, the combined therapy (BAPN and PD-1) is necessary to expand the percentage of GrzmB CD8^+^ T cells and the ratio of CD8^+^ to Treg cells and is associated with an increase in cytokine production. The combined treatment also leads to a decrease in the tumor sizes. Although these results are convincing as they are, confirmation of the results in another model would strengthen the results.

*Reviewer #2:*

In this manuscript, the authors provide a thorough analysis of the ECM architecture and stiffness in 4 murine tumor models. They then attempt to correlate ECM architecture and mechanics with T-cell migration and PD-1 efficacy. Substantive concerns are as follows:

1. The study is highly correlative with inadequate sample size to be conclusive. The authors attempts to draw conclusions about when stiffness does and doesn't affect migration by attempting to interpret data across 4 very different tumor types. In two tumors the migration changes with BAPN and with 2 it does not. It is not possible to draw a conclusion based on 2 points.

2. Data regarding the relationship between collagen organization and stiffness has been reported previously (as cited by the authors).

3. Sirius Red staining is referred to and described in the text but no images are shown. Likewise, no SWE images are provided to show the relative heterogeneity described in the text. This is important since so much of the conclusions rests on this data.

4. The Results section discussing figure 1 emphasizes heterogeneity in stiffness, however none of the data shown depict spatial stiffness heterogeneities.

5. The rationale for the choice of cancer models is not clear.

6. Why is mPDAC measured and reported differently in figure 2A than the other tumor types?

7. Why is 40kPa chosen as the cut-off for "stiff?"

8. Mean-squared displacement is the more appropriate metric to describe cell path (and more conventional) rather than "straightness"

9. How many cells were studied for each parameter in each condition in Table 2?

10. The authors study migration of cells on slices, but isn't the more appropriate metric to study cell invasion into the tissue?

*Reviewer #3:*

The focus of the manuscript by Nicolas-Boluda et al. is timely as it has been shown by this team and by others that dense collagen fibers and other features of the matrix architecture surrounding tumors may form a barrier for T cell infiltration into solid tumors. Despite the authors' claims, however, the data in this manuscript fall short of definitively demonstrating that response to anti-PD-1 therapy and T cell migration into tumors is improved upon reduction of collagen cross-linking. I have a number of concerns that would require additional substantive experiments to be adequately addressed. Below I list major points that should be addressed before further consideration for publications.

1) BAPN is used as a covalent inhibitor of LOX activity however the authors provide no evidence that the drug is having the expected effects in vivo. In order to draw specific conclusions about these studies the authors would need to provide measurements of collagen cross-links (DHLNL, PYP, DPD).

2) Imbalance between the mechanical characterization of multiple tumor models with little space for defining the effect of tumor stiffness on anti-PD-1 efficacy and T cell distribution, motility and activation.

3) Rationale for selected tumor models relative to human tumors was unclear.

4) Sample sizes, # independent experiments and statistical analyses were inadequate across multiple figures.

5) Measurements of stiffness, collagen structure and T cell speed should be provided for all treatment conditions (control, LOXi, PD1i and combo) rather than just for LOX inhibition.

6) Lox inhibition was performed in a preventive setting. Do the authors think LOX inhibition would be as effective in changing tumor stiffness and matrix architecture if the treatment started at the same time point as anti-PD-1?

7) In Figure 1 the correlation of tissue stiffness/collagen accumulation with tumor volume in clinical samples should be provided in order to attribute collagen cross-linking to tumor progression.

8) The efficacy data in Figure 6 should be accompanied by survival data.

---

## [Author Response]

Reviewer #1:[…] In the first part of their study, the authors analyze the structure heterogeneity of 5 different carcinomas, i.e. subcutaneous model of cholangiocarcinoma (EGI-1), subcutaneous (MET-1) and transgenic model (MMTV-PyMT) of mouse breast carcinoma, orthotopic (mPDAC) and subcutaneous (KPC) models of mouse pancreatic ductal adenocarcinoma.They measure the tumor stiffness during tumor growth using Shear Wave Elastography (SWE) and analyze the organization of the collagen fibers in these models. To my knowledge, this represents the first characterization of different tumor models classically used to study tumor immunity and is thus very useful for the scientific community. In particular, the authors show a correlation between high tumor stiffness and accumulation of thick and densely packed collagen fibers.Minor modifications: The authors should indicate more clearly the number of mice and tumors investigated.

Following the Reviewer’s request, we have now clearly stated, in the figure legends, the number of mice and tumors investigated for each experiment.

In the second part of their study, the authors treat the mice with beta-aminopropionitrile (BAPN), an inhibitor for LOX enzymatic activity in the drinking water and analyze the stiffness of tumors and collagen fiber organization in tumors. They show the heterogeneity of response in the different models in both stiffness modulation and collagen fibers remodeling. Mostly this treatment reduces the stiffness of tumors without affecting their growth.Minor modifications: The authors should clarify how "normalized tumor stiffness" indicated in the legend of figure 2 is calculated. Indeed, this is an important point since tumor stiffness is associated to the sizes of tumors. Moreover, they should also indicate more clearly the number of mice and tumors investigated. Concerning collagen fibers orientation, authors should use a dot plot representation instead of bar histograms in order to show the distribution in the different tumors.

The tumor stiffness measured at each time point was normalized to that of the first time point. Thereby, values represent the increase of tumor stiffness in time regardless of the value of the first measurement. This has been indicated in the supplementary materials, (shear wave elastography section) as follows:

“The normalized tumor stiffness was calculated by normalizing each time point measurement to that of the initial time point (Stiffness t_n_/ Stiffness t_0_).”

As previously indicated the number of mice/tumors analyzed has been added to the legend of Figure 2.

Concerning the collagen fiber orientation, this measurement is already represented as a dot plot (Figure 3B), whilst Figure 3C shows a representative example of fiber orientation distribution.

The authors then analyze how BAPN treatment modifies the migration of T lymphocytes in the tumors. Because of the different models used, the authors either added activated purified T cells from human donors (EGI-1model), or mouse activated T cells (MMTV-PyMT tumor model) or followed the motility of human resident T cells in mPDAC and KPC mice tumor models. Although the models are very different, the correlation between tumor stiffness and T cell speed and T cell displacement is especially striking in tumors from BAPN treated mice. It seems that T cell motility responds to two different regimens in tumor from untreated or BAPN treated mice. This might be due to difference of stiffness in untreated and treated mice but might also results from another parameter.Minor modifications: The authors should discuss this point. Indeed, the main conclusion of their work and short title of their study is that the main parameter involved in T cell motility and access to the tumor is tumor stiffness but then the slopes should be the same as in the spontaneous MMTV-PyMT tumor model. There are probably other parameters involved in the regulation.

We agree with the Reviewer that along with tumor stiffness other determinants can also control the migration of T cells in tumors. These include chemical factors (e.g., chemokines) but also cellular entities that could either positively or negatively regulate T cell intratumoral migration. Although the anti-LOX strategy that we used in our study targets collagen crosslinking, we cannot rule out possible indirect effects of LOX inhibition on cells and elements important for the migration of T cells. Thus, we have added a chapter in the discussion highlighting the fact that tumor ECM is part of a microenvironment having detrimental effects on T cell migration.

*“*In this study, we did not take into consideration possible effects of LOX inhibition on other determinants that could either positively or negatively regulate T cell migration. […] The data in Figure 6 showing a decrease in PMN and Treg and an increase in MHC class II+ macrophages upon BAPN treatment support this assumption”.

The authors then investigate the effect of BAPN treatment of tumor bearing mice on response to PD-1 immunotherapy. They perform experiments in KPC tumor bearing mice and show that BAPN treatment alone significantly decreases the number of neutrophils, increases the presence of MHCII+ TAMs. Yet, the combined therapy (BAPN and PD-1) is necessary to expand the percentage of GrzmB CD8^+^ T cells and the ratio of CD8^+^ to Treg cells and is associated with an increase in cytokine production. The combined treatment also leads to a decrease in the tumor sizes. Although these results are convincing as they are, confirmation of the results in another model would strengthen the results.

We agree. However, finding a relevant tumor model to perform combination treatments (immune checkpoint and collagen crosslinking inhibitors) has proven to be difficult. In principle, such a model should exhibit three main features, namely a stiff tumor, ECM normalization upon LOX inhibition, and T cells infiltrated in tumors. The KPC tumor model that we have used for these combination experiments encompasses all three characteristics. However, this is not the case with the other tumor models that we have studied in the first part of our study. Specifically, the MET-1 model is not stiff and thus does not respond to LOX inhibition in terms of ECM structure. On the other hand, spontaneous tumor models are structurally relevant but often lack the presence of T cells in tumors as in the case of MMTV-PyMT tumors.

Nonetheless, we performed additional experiments in three other tumor models that gave us inconclusive results. We provide here the results of experiments that we have conducted in transplanted MMTV-PyMT tumors. This model consists of injecting subcutaneously cells from freshly dissociated MMTV-PyMT tumors (5). In this condition, the tumors grow much faster than in the spontaneous model. The difference in development pace results in differences at the level of the tumor composition. The extracellular matrix in the spontaneous tumors is much more abundant than in the transplanted ones (Author response image 1). This translates also at the macroscopic level with spontaneous tumors being stiffer than transplanted tumors at equivalent tumor volumes (Author response image 1). Despite the differences in the extracellular matrix density, we decided to test the combination of anti-PD1 and anti-CTLA4 with LOX inhibition.

**Author response image 1. respfig1:** (A) Confocal microscopy images of transplanted and spontaneous MMTV-PyMT tumors. Fibronectin (red), EpCAM (tumor cells, blue) CD8 (green). (B) Correlation of tumor mean stiffness and tumor volume in spontaneous and transplanted MMTV-PyMT tumors.

Mice with transplanted MMTV-PyMT tumors respond well to anti-CTLA and anti-PD-1 as evidenced by a significant decrease in tumor volume (Author response image 2). However, no further decrease is observed when immune checkpoint inhibitors are combined with BAPN. We believe that the lack of additional effect is related to the fact that LOX inhibition does not produce a significant decrease in tumor stiffness as it is the case in other models (Author response image 2). In addition, when we examined the presence of CD8^+^ T cells in the tumor, we could see that both the immunotherapies alone and BAPN alone separately induce an increase in CD8 T cells in the tumor (compared to the control), however, the combination does not enhance this accumulation (Author response image 2).

**Author response image 2. respfig2:** Tumor fold change after combination therapy of BAPN and anti-PD1/anti-CTLA 4 in transplanted MMTV-PyMT tumor. (B) Normalized tumor stiffness (C) %CD8^+^T cell/CD45+ cells (D) Confocal microscopy images Fibronectin (red), EpCAM (tumor cells, blue) CD8 (green).

In conclusion, we are convinced that finding another relevant tumor model to perform combination treatments is important. However, it will require considerable efforts that would go beyond the scope of the current study and should be addressed in future work. What seems to us important is that the variety of models investigated in the paper reveals the most relevant and intertwined parameters, namely tumor architecture (segregation of ECM with respect to tumor islet) collagen structure, tumor stiffness and T cell infiltration that conditions the response to LOX inhibition and immune checkpoint inhibitors.

Reviewer #2:In this manuscript, the authors provide a thorough analysis of the ECM architecture and stiffness in 4 murine tumor models. They then attempt to correlate ECM architecture and mechanics with T-cell migration and PD-1 efficacy. Substantive concerns are as follows:1. The study is highly correlative with inadequate sample size to be conclusive. The authors attempts to draw conclusions about when stiffness does and doesn't affect migration by attempting to interpret data across 4 very different tumor types. In two tumors the migration changes with BAPN and with 2 it does not. It is not possible to draw a conclusion based on 2 points.

We acknowledge that, although BAPN resulted in an overall increase in T cell migration in every tested model, not all tumor models react similarly in terms of motility parameters (Figure 5). However, results from additional experiments that we performed in the KPC model pooled with prior results clearly show enhancements of T cell displacements upon BAPN treatment in this model too. This adds key new information to the first version of our manuscript and strengthens the conclusion on the impact of BAPN treatment on T cell migration. Overall, LOX inhibition increases T cell displacements within tumors from all tumor models and produces variable effects in the other parameters studied (velocity and straightness).

Of note, results from Figure 5 were obtained with data pooled from all mice, either treated or not with BAPN. We also wish to point out that we extended our analysis at the level of the individual mouse in every tumor model (Figure 6). The data in this figure show a clear inverse linear correlation between T cell migration speed and mean tumor stiffness. This analysis allowed us to demonstrate a marked difference in intratumoral T cell migration when measured in control or BAPN-treated conditions.

2. Data regarding the relationship between collagen organization and stiffness has been reported previously (as cited by the authors).

Indeed, such a relationship between tumor stiffness and collagen organization has been reported previously. However, we provide in our study novel insights about this relationship through a multiscale analysis performed in several preclinical mouse models of pancreatic, breast, and bile duct carcinomas, presenting different ECM organizations which to our knowledge has never been attempted so far.

This has been added to the discussion:

“Previous studies had already explored the link between tumor stiffness and collagen architecture (6-8). Here, we provide novel insights about this relationship through a multiscale analysis performed in several preclinical mouse models of pancreatic, breast and bile duct carcinomas, presenting different ECM organization and thus covering the heterogeneity found in cancer patients”.

3. Sirius Red staining is referred to and described in the text but no images are shown. Likewise, no SWE images are provided to show the relative heterogeneity described in the text. This is important since so much of the conclusions rests on this data.

Figure 3—figure supplement 1 shows representative Sirius red images for each of the tumors. Shear Wave Elastography images are shown in Figure 2 as well as in Figure 2—figure supplement 2-6.

4. The Results section discussing figure 1 emphasizes heterogeneity in stiffness, however none of the data shown depict spatial stiffness heterogeneities.

Figure 2C clearly illustrates the difference in heterogeneity in stiffness within the tumor. For example in the case of EGI-1, the control tumor shows large heterogeneity with stiffness values ranging from 1 to over 40 kPa. On the other hand, the BAPN treated tumors show less heterogeneity with values ranging from 1 to 10-12 kPa.

5. The rationale for the choice of cancer models is not clear.

We agree with the reviewer’s comment and now discuss the different tumor models used in our study more explicitly, as follows (p16):

“In this study, we have set up and characterized five different solid tumor models covering three types of carcinomas that differentiate in terms of tumor-stroma organization, tumor stiffness, collagen structure, and T cell infiltration. […] On the other hand, the MET-1 model only contains thin stromal compartments and unlike the previous models, tumors are not very stiff, probably reflecting a specific subtype of human breast cancers”.

6. Why is mPDAC measured and reported differently in figure 2A than the other tumor types?

Experiments on the mPDAC tumor model have been performed by our collaborators, M. Ponzo and I. Cascone, who have established this model. Orthotopic tumor cell implantation and housing of tumor-bearing mice were carried out in their lab not equipped with a shear wave elastography apparatus. Hence, mice had to be transported to the small animal imaging facility in order to assess tumor stiffness. For safety and ethical reasons as well as to prevent animal stress during the transportation that may have biased the experiment, mice were only transported once at the end of the experiments to perform these measurements.

7. Why is 40kPa chosen as the cut-off for "stiff?"

This cut-off was chosen based on the results from a previous study performed by our group (10) which is now cited in the revised version of our manuscript.

8. Mean-squared displacement is the more appropriate metric to describe cell path (and more conventional) rather than "straightness"

Indeed, mean-squared displacement and straightness are two locomotion parameters that do not provide the same information. Mean-squared displacement gives information about the type of migration involved (random, directed, or confined). The straightness, on the other hand, indicates the type of trajectory the cell accomplishes, whether it is straight or curvy.

For some experiments, we have analyzed the mean-squared displacement of T cells in control and BAPN-treated conditions. Although there is a tendency for T cells to show higher mean-squared displacements in BAPN conditions as compared to control conditions, no statistical difference was noticed.

We feel our manuscript already contains many motility parameters. However, we will be willing to include Mean-squared displacement data should the reviewers and editors support their inclusion.

9. How many cells were studied for each parameter in each condition in Table 2?

The number of T cells analyzed in each slice was not equal, there were between 70 and 250 T cells in each slice analyzed. These numbers have now been included in the legend of Supplementary File 4 (former Table 2).

10. The authors study migration of cells on slices, but isn't the more appropriate metric to study cell invasion into the tissue?

In principle, intravital two-photon microscopy is the method of choice to monitor T cell migration in intact tissues. However, this approach is technically challenging to set up with some tumors and especially with the orthotopic tumor model used here. Instead, we have used a technique of fresh tumor slices that has been instrumental for our team in elucidating the mechanisms of T cell migration within tumors (11,12). The fresh slices preserve the complex multicellular tumor microenvironment and they are thick enough to study T cell migration within the tissue.

We now provide details in the Results section about the slice assay as follows:

“The tumor slice assay that we have established preserves the original tissue microenvironment and permits to monitor with confocal microscopy the behavior of either ex vivo purified and plated T cells or endogenous T cells labeled with directly-coupled fluorescent antibodies”.

Reviewer #3:The focus of the manuscript by Nicolas-Boluda et al. is timely as it has been shown by this team and by others that dense collagen fibers and other features of the matrix architecture surrounding tumors may form a barrier for T cell infiltration into solid tumors. Despite the authors' claims, however, the data in this manuscript fall short of definitively demonstrating that response to anti-PD-1 therapy and T cell migration into tumors is improved upon reduction of collagen cross-linking. I have a number of concerns that would require additional substantive experiments to be adequately addressed. Below I list major points that should be addressed before further consideration for publications.1) BAPN is used as a covalent inhibitor of LOX activity however the authors provide no evidence that the drug is having the expected effects in vivo. In order to draw specific conclusions about these studies the authors would need to provide measurements of collagen cross-links (DHLNL, PYP, DPD).

Traditional methods to measure fibrillar collagen crosslinks include high-performance liquid chromatography (HPLC) and liquid chromatography-mass spectrometry (LC–MS). A systematic search of the MEDLINE database indicates that the number of labs in the world able to perform such measurement is limited. In fact, none of the proteomic facilities we contacted was having the proper expertise to conduct collagen crosslinks analysis. As an alternative approach, we have decided to exploit the intrinsic fluorescence of two trivalent LOX-generated crosslinks, pyridinoline (PYD) and deoxypyridinoline (DPD), (Ex 325 nm, Em 400 nm) (13) and use two-photon imaging to measure the natural fluorescence of PYD and DPD in tumor sections from mice treated or not with BAPN. Notably, such a fluorescence imaging strategy to assess the level of collagen crosslinks in tissue sections has already been conducted and validated by comparing microscopy results to those obtained with traditional amino acid analyses (14).

In the revised version of our manuscript, we include new data (new Figure 2—figure supplement 1) showing that the average fluorescence signals of PYD and DPD, measured in SHG positive regions, are significantly decreased in BAPN as compared to control conditions. Changes were also made to the text to reflect these observations. This result together with the effects of BAPN treatment on collagen organization in tumors suggests that this pharmacological compound has the expected effects in vivo.

2) Imbalance between the mechanical characterization of multiple tumor models with little space for defining the effect of tumor stiffness on anti-PD-1 efficacy and T cell distribution, motility and activation.

We believe that the novelty of our study comes from a comprehensive investigation of tumor mechanical properties (particularly tumor stiffness which is a physical biomarker easily accessible in patients through clinical elastography) and their relationships with the ECM architecture and subsequent impacts on T cell intratumoral migration. In this paper, we have investigated 5 different preclinical models recapitulating the heterogeneity of the ECM architecture and representing distinct evolution of the tumor stiffness upon tumor progression, with important consequences on T cell infiltration and migration. In this regard, LOX inhibition was used and characterized as a critical modulator of the collagen architecture, with profound consequences on tumor stiffness and T cell migration, acting differently depending on the tumor type. Our main message is thus that the stiffness of the tumor, and more generally the precise architecture of the tumor, must be carefully examined in order to categorize patients that could be resistant to immunotherapies due to poor access and migration capacity of effector T cells. The use of anti-PD-1 in combination with LOX inhibition was proposed in our study as an illustration of this concept in one of the preclinical models, knowing this combined strategy is not enough to induce complete tumor regression. Nevertheless and as requested by the Reviewer in his/her point #5, we provide novel data about the effects of LOX inhibition and PD-1 blockade on tumor stiffness, stroma organization, and T cell motility in KPC tumors (new Figure 7—figure supplement 1).

3) Rationale for selected tumor models relative to human tumors was unclear.

We now discuss the different tumor models used in our study more explicitly, as follows (p16):

“In this study, we have set up and characterized five different solid tumor models covering 3 types of carcinomas that differentiate in terms of tumor-stroma organization, tumor stiffness, collagen structure, and T cell infiltration. […] On the other hand, the MET-1 model only contains thin stromal compartments and unlike the previous models, tumors are not very stiff, probably reflecting a specific subtype of human breast cancers”.

4) Sample sizes, # independent experiments and statistical analyses were inadequate across multiple figures.

Following the Reviewer’s request, we have now clearly stated, in the figure legends, the number of mice and tumors investigated for each experiment. The number of T cells analyzed for each motility experiment (Supplementary File 4) was also included.

5) Measurements of stiffness, collagen structure and T cell speed should be provided for all treatment conditions (control, LOXi, PD1i and combo) rather than just for LOX inhibition.

Following the Reviewer’s suggestions, we performed additional experiments measuring tumor stiffness, collagen structure, and T cell motility within KPC tumors for all treatment conditions (Control, BAPN, anti-PD1 and combination). Our new results (new Figure 7—figure supplement 1) indicate that anti-PD1 treatment does not affect tumor stiffness nor collagen organization. However, we found that PD-1 blockade results in an increase in T cell motility as comparable to that induced by BAPN treatment. Changes were also made to the text to reflect these observations. We also discuss these findings as follows:

“Along with tumor ECM, other cells and factors play important roles in controlling T cell migration. […] It is well established that anti-PD-1 treatment is associated with the production of IFNγ and inflammatory chemokines (e.g., CXCL10) that are presumably responsible for enhancing T cell motility in tumors (1)”.

6) Lox inhibition was performed in a preventive setting. Do the authors think LOX inhibition would be as effective in changing tumor stiffness and matrix architecture if the treatment started at the same time point as anti-PD-1?

In response to this important point, we have included several sentences to the discussion as follows:

“In this study, LOX inhibition was performed in a preventive setting. […] For example, blocking LOX in combination with gemcitabine reduced metastases and increased survival of the mice when treatment was started in the early stages of the disease but not at later stages (16)”.

7) In Figure 1 the correlation of tissue stiffness/collagen accumulation with tumor volume in clinical samples should be provided in order to attribute collagen cross-linking to tumor progression.

We have included in our revised manuscript references of clinical studies showing a correlation between tumor stiffness and tumor size. It reads as follows:

“In human breast cancer, a significant correlation between tumor stiffness and tumor size was demonstrated (17,18). […] In addition, we show a correlation between tumor stiffness measured non-invasively with collagen accumulation associated with a segregated architecture of thick and densely packed collagen fibers (Sirius red positive) surrounding tumor nests”.

8) The efficacy data in Figure 6 should be accompanied by survival data.

Such experiments are not possible in this tumor model. Subcutaneous KPC tumors do not metastasize and therefore mice do not die from even very large tumors. For ethical reasons, mice must be killed when tumors reached 1.5 cm^3^.

References :

1. Peng W, Liu C, Xu C, Lou Y, Chen J, Yang Y, et al. PD-1 blockade enhances T-cell migration to tumors by elevating IFN-γ inducible chemokines. Cancer Res 2012;72(20):5209-18 doi 10.1158/0008-5472.CAN-12-1187.

2. Maller O, Drain AP, Barrett AS, Borgquist S, Ruffell B, Zakharevich I, et al. Tumour-associated macrophages drive stromal cell-dependent collagen crosslinking and stiffening to promote breast cancer aggression. Nat Mater 2021;20(4):548-59 doi 10.1038/s41563-020-00849-5.

3. Deligne C, Murdamoothoo D, Gammage AN, Gschwandtner M, Erne W, Loustau T, et al. Matrix-targeting immunotherapy controls tumor growth and spread by switching macrophage phenotype. Cancer Immunology Research 2020:canimm.0276.2019 doi 10.1158/2326-6066.CIR-19-0276.

4. Manaster Y, Shipony Z, Hutzler A, Kolesnikov M, Avivi C, Shalmon B, et al. Reduced CTL motility and activity in avascular tumor areas. Cancer Immunol Immunother 2019;68(8):1287-301 doi 10.1007/s00262-019-02361-5.

5. Weiss JM, Guerin MV, Regnier F, Renault G, Galy-Fauroux I, Vimeux L, et al. The STING agonist DMXAA triggers a cooperation between T lymphocytes and myeloid cells that leads to tumor regression. Oncoimmunology 2017;6(10):e1346765 doi 10.1080/2162402X.2017.1346765.

6. Samani A, Zubovits J, Plewes D. Elastic moduli of normal and pathological human breast tissues: an inversion-technique-based investigation of 169 samples. Phys Med Biol 2007;52(6):1565-76 doi 10.1088/0031-9155/52/6/002.

7. Venkatesh SK, Yin M, Glockner JF, Takahashi N, Araoz PA, Talwalkar JA, et al. MR elastography of liver tumors: preliminary results. AJR Am J Roentgenol 2008;190(6):1534-40 doi 10.2214/AJR.07.3123.

8. Mieulet V, Garnier C, Kieffer Y, Guilbert T, Nemati F, Marangoni E, et al. Stiffness increases with myofibroblast content and collagen density in mesenchymal high grade serous ovarian cancer. Sci Rep 2021;11(1):4219 doi 10.1038/s41598-021-83685-0.

9. Lin EY, Jones JG, Li P, Zhu L, Whitney KD, Muller WJ, et al. Progression to malignancy in the polyoma middle T oncoprotein mouse breast cancer model provides a reliable model for human diseases. Am J Pathol 2003;163(5):2113-26 doi 10.1016/s0002-9440(10)63568-7.

10. Marangon I, Silva AA, Guilbert T, Kolosnjaj-Tabi J, Marchiol C, Natkhunarajah S, et al. Tumor Stiffening, a Key Determinant of Tumor Progression, is Reversed by Nanomaterial-Induced Photothermal Therapy. Theranostics 2017;7(2):329-43 doi 10.7150/thno.17574.

11. Peranzoni E, Lemoine J, Vimeux L, Feuillet V, Barrin S, Kantari-Mimoun C, et al. Macrophages impede CD8 T cells from reaching tumor cells and limit the efficacy of anti-PD-1 treatment. Proceedings of the National Academy of Sciences of the United States of America 2018;115(17):E4041-E50 doi 10.1073/pnas.1720948115.

12. Salmon H, Franciszkiewicz K, Damotte D, Dieu-Nosjean MC, Validire P, Trautmann A, et al. Matrix architecture defines the preferential localization and migration of T cells into the stroma of human lung tumors. J Clin Invest 2012;122(3):899-910 doi 10.1172/jci45817.

13. Richards-Kortum R, Sevick-Muraca E. Quantitative optical spectroscopy for tissue diagnosis. Annu Rev Phys Chem 1996;47:555-606 doi 10.1146/annurev.physchem.47.1.555.

14. Marturano JE, Xylas JF, Sridharan GV, Georgakoudi I, Kuo CK. Lysyl oxidase-mediated collagen crosslinks may be assessed as markers of functional properties of tendon tissue formation. Acta Biomater 2014;10(3):1370-9 doi 10.1016/j.actbio.2013.11.024.

15. Nilsson M, Adamo H, Bergh A, Halin Bergstrom S. Inhibition of Lysyl Oxidase and Lysyl Oxidase-Like Enzymes Has Tumour-Promoting and Tumour-Suppressing Roles in Experimental Prostate Cancer. Sci Rep 2016;6:19608 doi 10.1038/srep19608.

16. Miller BW, Morton JP, Pinese M, Saturno G, Jamieson NB, McGhee E, et al. Targeting the LOX/hypoxia axis reverses many of the features that make pancreatic cancer deadly: inhibition of LOX abrogates metastasis and enhances drug efficacy. EMBO Mol Med 2015;7(8):1063-76 doi 10.15252/emmm.201404827.

17. Evans A, Whelehan P, Thomson K, McLean D, Brauer K, Purdie C, et al. Invasive breast cancer: relationship between shear-wave elastographic findings and histologic prognostic factors. Radiology 2012;263(3):673-7 doi 10.1148/radiol.12111317.

18. Song EJ, Sohn YM, Seo M. Tumor stiffness measured by quantitative and qualitative shear wave elastography of breast cancer. Br J Radiol 2018;91(1086):20170830 doi 10.1259/bjr.20170830.

19. Guerin MV, Regnier F, Feuillet V, Vimeux L, Weiss JM, Bismuth G, et al. TGFbeta blocks IFNalpha/β release and tumor rejection in spontaneous mammary tumors. Nat Commun 2019;10(1):4131 doi 10.1038/s41467-019-11998-w.